# Angiomotin functions in HIV-1 assembly and budding

Gaelle Mercenne, Steven L Alam, Jun Arii, Matthew S Lalonde, Wesley I Sundquist*

Department of Biochemistry, University of Utah, Salt Lake City, United States

**Abstract** Many retroviral Gag proteins contain PPXY late assembly domain motifs that recruit proteins of the NEDD4 E3 ubiquitin ligase family to facilitate virus release. Overexpression of NEDD4L can also stimulate HIV-1 release but in this case the Gag protein lacks a PPXY motif, suggesting that NEDD4L may function through an adaptor protein. Here, we demonstrate that the cellular protein Angiomotin (AMOT) can bind both NEDD4L and HIV-1 Gag. HIV-1 release and infectivity are stimulated by AMOT overexpression and inhibited by AMOT depletion, whereas AMOT mutants that cannot bind NEDD4L cannot function in virus release. Electron microscopic analyses revealed that in the absence of AMOT assembling Gag molecules fail to form a fully spherical enveloped particle. Our experiments indicate that AMOT and other motin family members function together with NEDD4L to help complete immature virion assembly prior to ESCRT-mediated virus budding.

## Introduction

As obligate parasites, viruses must harness host cellular pathways to complete many different steps in viral replication. For example, a large number of enveloped viruses, including HIV-1, usurp the cellular ESCRT (Endosomal Sorting Complexes Required for Transport) pathway to bud from cells (for recent reviews, see *Meng and Lever, 2013*; *Votteler and Sundquist, 2013*; *Weissenhorn et al., 2013*). Viral structural proteins typically use one or more short peptide motifs, termed 'late assembly domains', to bind and recruit early-acting ESCRT factors or ESCRT-associated E3 ubiquitin ligases to sites of viral assembly and budding. The three best characterized viral late assembly domains are: (1) the 'PTAP' motif (Pro-Thr/Ser-Ala-Pro), which binds the UEV domain of the TSG101 subunit of the ESCRT-I complex, (2) the 'YPX$_n$L' motif (Tyr-Pro-X$_n$-Leu, where X can vary in identity and sequence length), which binds the V domain of the ESCRT factor ALIX, and (3) the 'PPXY' motif (Pro-Pro-X-Tyr), which binds the WW domains of NEDD4 family E3 ubiquitin ligases. All three of these viral late assembly domains mimic cellular protein–protein interactions that facilitate ESCRT-dependent vesiculation processes such as multi-vesicular body (MVB) biogenesis and ectosome formation.

In addition to the three well-characterized late assembly domains described above, there are indications that enveloped viruses can also connect to the ESCRT pathway via alternative interactions that are not yet well-understood. For example, efficient release of the model paramyxovirus, PIV5, requires a non-canonical 'FPIV' late assembly domain motif, ubiquitylation of the matrix (M) protein, and host Angiomotin-like protein 1 (AMOTL1) (*Schmitt et al., 2005*; *Pei et al., 2010*; *Harrison et al., 2012*). There is no evidence that AMOTL1 can bind the FPIV late assembly domain, however, and any relationships between ubiquitin, AMOTL1 and the FPIV late assembly domain activity are not yet understood (*Pei et al., 2010*).

Owing to its medical importance, HIV-1 has become a leading model for understanding ESCRT pathway recruitment and viral budding. The C-terminal p6 region of the viral Gag structural protein contains canonical PTAP and YPX$_n$L motifs that serve as the major functional late assembly domains in most cell types. Nevertheless, the release of 'crippled' HIV-1 Gag constructs that lack both of these

*For correspondence: wes@biochem.utah.edu

**Competing interests:** The authors declare that no competing interests exist.

**eLife digest** To multiply and spread infections, viruses must enter and exit cells. Once inside a cell, many viruses conscript the cell's machinery to produce new viral particles and release them into the surroundings. Some viruses—like HIV-1—exit the cell in a way that leads to them being wrapped (or 'enveloped') in membrane from the host cell.

A virus protein called Gag is required for the release of HIV-1 and other enveloped viruses. In some cases, Gag proteins bind directly to members of the NEDD4 protein family to enable the viruses to be released. However, the Gag protein from HIV-1 does not appear to be able to interact directly with NEDD4 proteins, so it was not clear how Gag works in this case.

Mercenne et al. studied how HIV-1 is released from human cells grown in the laboratory. The experiments show that members of a human protein family called the Angiomotins bind to both Gag and NEDD4L (a member of the NEDD4 family) and are required for the efficient release of viruses. Using a technique called electron microscopy, Mercenne et al. observed that when Angiomotins are present, Gag proteins assemble in spheres at the cell membrane and viruses are able to exit the cell. However, when Angiomotins are depleted or absent, incomplete spheres of Gag proteins accumulate on the inner membrane surface and viruses are not released.

These findings show that Angiomotins act as a link between Gag and NEDD4L to promote the release of HIV-1 from human cells. The next step will be to learn precisely how this works. There are indications that the Angiomotins may also be involved in the release of other enveloped viruses, so the findings may be useful for the development of treatments for a variety of viral infections.

late domains can still be stimulated by overexpression of the HECT E3 ubiquitin ligase, NEDD4L (also known as NEDD4-2) (*Chung et al., 2008*; *Usami et al., 2008*). Humans express nine different NEDD4 protein family members (reviewed in *Ingham et al., 2004*; *Bernassola et al., 2008*), but NEDD4L stimulates virus budding more potently than any other NEDD4 family member, implying that HIV-1 Gag preferentially engages this E3 ubiquitin ligase (*Chung et al., 2008*; *Usami et al., 2008*).

Previous studies have defined many of the requirements for NEDD4L-stimulated HIV-1 release. NEDD4L has many different isoforms (*Chen et al., 2001*; *Dunn et al., 2002*; *Itani et al., 2005*), but isoform 2 (also called NEDD4-2s, and here termed NEDD4L) potently stimulates the budding of crippled HIV-1 constructs, and uniquely rescues the Gag processing defects that accompany defective budding (*Chung et al., 2008*; *Usami et al., 2008*). This NEDD4L isoform contains only the final 31 residues of the N-terminal C2 membrane binding domain, a central linker region that contains four WW domains, and a C-terminal HECT E3 ubiquitin ligase domain that forms Lys-63-linked poly-ubiquitin chains (see *Figure 1A*) (*Kim and Huibregtse, 2009*; *Weiss et al., 2010*). Stimulation of HIV-1 budding requires NEDD4L E3 ubiquitin ligase activity, indicating that Lys-63-linked poly-ubiquitin chains perform an essential function (*Chung et al., 2008*; *Usami et al., 2008*). The functional target(s) of Lys-63 poly-ubiquitylation has not been established definitively, but HIV-1 Gag itself is the leading candidate because functional virus budding correlates with Lys-63-linked poly-ubiquitylation of Gag, and because NEDD4L truncation constructs that include the HECT domain can be functionally rescued by fusion to the HIV-1 Gag-targeting cyclophilin A domain (*Weiss et al., 2010*; *Sette et al., 2013*). HIV-1 Gag lacks identifiable PPXY motifs, however, and there is no evidence that Gag can bind NEDD4L directly.

These observations imply that NEDD4L may be recruited to sites of HIV-1 assembly through an alternative mechanism. We undertook the present study with the goal of identifying co-factors that might help link NEDD4L to HIV-1 Gag and stimulate virus budding. We identified Angiomotin (AMOT) as a protein that binds both NEDD4L and HIV-1 Gag, and functions in HIV-1 assembly and release. AMOT is required to complete virion assembly and envelopment, and is therefore a new host factor that acts in a previously uncharacterized step of HIV-1 morphogenesis.

## Results

### NEDD4L binds Angiomotin

Affinity purification/mass spectrometry experiments were performed to identify candidate NEDD4L binding partners. A number of cellular proteins co-purified with OSF (One-STrEP-FLAG)-tagged

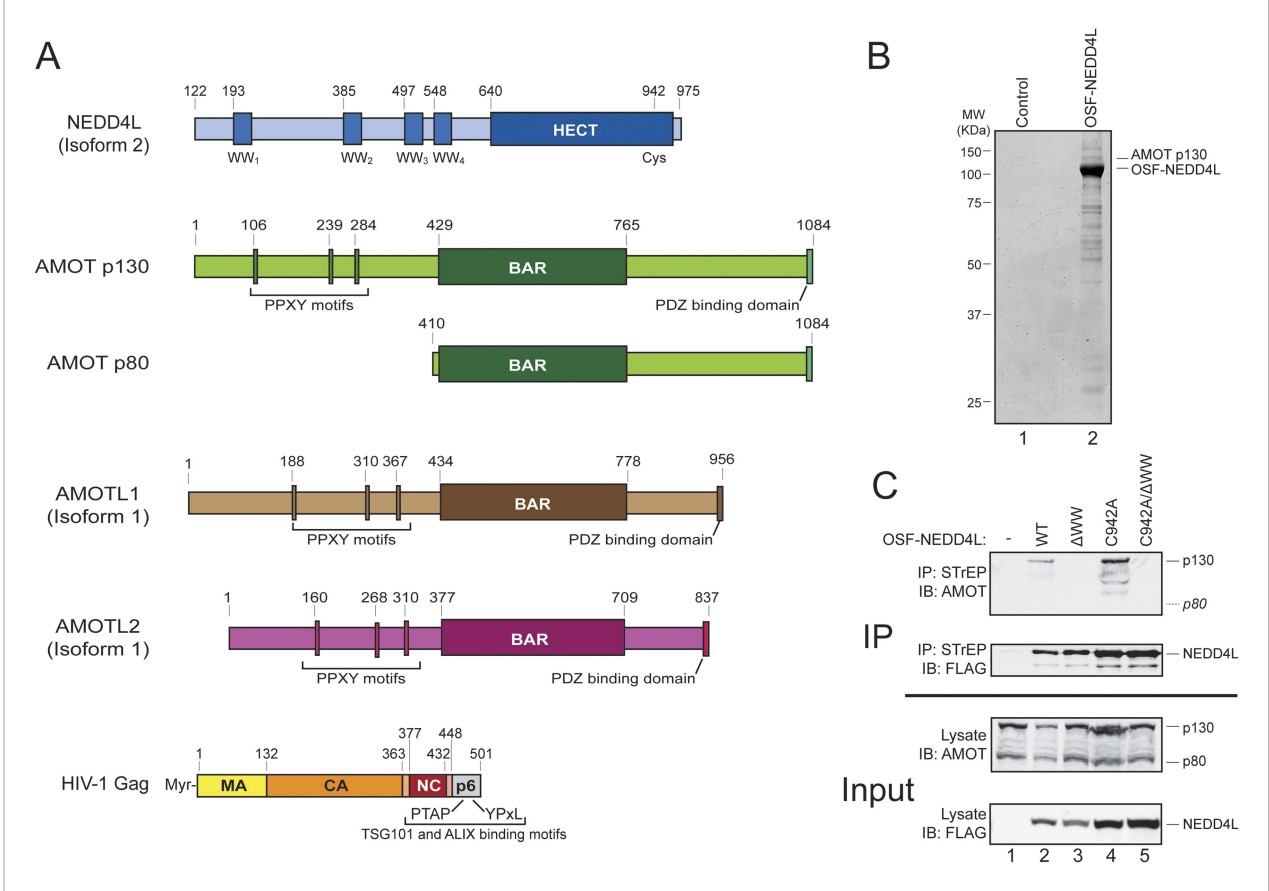

**Figure 1**. AMOT p130 is a NEDD4L binding partner. (**A**) Schematic illustrations of the domain structures and motifs in NEDD4L, AMOT p130 and p80, AMOTL1, AMOTL2 and HIV-1 Gag. Here and throughout, NEDD4L refers to a naturally occurring protein isoform (isoform 2) that contains only the C-terminal 32 residues of the C2 domain (Genebank accession number AAM46201.1, denoted NEDD4Ls in other publications and NEDD4L$_{\Delta C2}$ in our previous publication (*Chung et al., 2008*)). To maintain consistency with that publication, the NEDD4L numbering scheme used here corresponds to NEDD4L isoform 1, which is 121 residues longer at the N-terminus and contains the entire C2 domain (denoted NEDD4L$_{WT}$ in reference *Chung et al., 2008*). (**B**) Affinity co-purification and identification of AMOT p130 as a binding partner of OSF (One-STrEP-FLAG)-tagged NEDD4L. SDS-PAGE/ Coomassie-stained gel showing STrEP-Tactin matrix affinity purified proteins from 293T cells transfected with an OSF-NEDD4L expression construct (lane 2) or an empty vector control (lane 1). Labels denote OSF-NEDD4L (bait) and AMOT p130 (prey). Trypsin-digested AMOT p130 peptides identified by mass spectrometric analyses are summarized in *Figure 1—figure supplement 1*. A representative image from three independent repetitions is shown. (**C**) NEDD4L WW domains are required for AMOT p130 binding. Western blots showing co-immunoprecipitations of endogenous 293T cell AMOT proteins (prey) with OSF-NEDD4L (bait). Panel 1: endogenous AMOT proteins (anti-AMOT blot) co-immunoprecipitated with OSF-NEDD4L baits from cells that lacked exogenous OSF-NEDD4L (lane 1, control), expressed wild type OSF-NEDD4L (lane 2), expressed OSF-NEDD4L$_{\Delta WW}$ (lane 3, construct has inactivating point mutations in all four WW domains), expressed OSF-NEDD4L$_{C942A}$ (lane 4, construct has an inactivating point mutation in the HECT E3 domain), or expressed OSF-NEDD4L$_{C942A/\Delta WW}$ (lane 5, construct has inactivating point mutations in the WW and HECT E3 domains). Panel 2: same co-immunoprecipitation experiment blotted with anti-FLAG antibodies to detect OSF-NEDD4L proteins. Panel 3: input levels of endogenous AMOT (anti-AMOT blot, 10% of total). Panel 4: input levels of exogenous OSF-NEDD4L (anti-FLAG, 10% of total). Note that endogenous AMOT p80 was present in the input lysate, but did not co-immunoprecipitate with OSF-NEDD4L. Representative image from three independent repetitions.

The following figure supplements are available for figure 1:

**Figure supplement 1**. Identification of AMOT p130 as a NEDD4L binding partner.

**Figure supplement 2**. The AMOT p130 PPXY motifs contribute to NEDD4L binding.

NEDD4L, as visualized by SDS-PAGE (*Figure 1B*, compare lanes 1 and 2). Prominent co-purifying proteins were excised from the gel, digested with trypsin, and identified by mass spectrometry. The p130 isoform of human Angiomotin was unambiguously identified as a protein that co-purified with OSF-NEDD4L in three independent repetitions of this experiment, (lane 2, labeled AMOT

p130, peptide coverage data are summarized in *Figure 1—figure supplement 1*). Smaller AMOT p130 fragments were also identified, as was the related AMOT family protein, Angiomotin-like protein 1 (AMOTL1, data not shown).

Cells express two AMOT isoforms, a longer p130 isoform, and a shorter C-terminal p80 isoform that is expressed from an alternatively spliced message (*Figure 1A*) (*Moreau et al., 2005*). AMOT p130 binds NEDD4 protein family members via interactions between the four central NEDD4 WW domains and three PPXY motifs located within the N-terminal extension that is unique to the AMOT p130 isoform (*Figure 1A*) (*Wang et al., 2012*). In good agreement with this report and with our affinity purification/mass spectrometry results, we found that AMOT p130 co-immunoprecipitated with exogenously expressed OSF-NEDD4L (*Figure 1C*, panel 1, lane 2). In contrast, AMOT p130 did not co-immunoprecipitate with a mutant OSF-NEDD4L protein that contained inactivating Trp to Ala substitutions in the key PPXY-binding 'W39' residues from each of the four WW domains (denoted OSF-NEDD4L$_{\Delta WW}$, panel 1, compare lanes 2 and 3) (*Otte et al., 2003*).

A previous study has demonstrated that NEDD4 proteins can ubiquitylate and reduce AMOT p130 levels (*Wang et al., 2012*), and we also observed that OSF-NEDD4L overexpression modestly reduced AMOT p130 levels (*Figure 1C*, panel 3, compare lanes 1 and 2). We therefore tested whether AMOT p130 also co-precipitated with OSF-NEDD4L proteins that carried inactivating Cys942Ala point mutations in the active site cysteine of the HECT E3 ubiquitin ligase domain (OSF-NEDD4L$_{C942A}$). This mutation increased OSF-NEDD4L levels in the lysate and immunoprecipitate (panels 2 and 4 compare lanes 4 and 5 to lanes 2 and 3) and restored normal AMOT protein levels (panel 3, compare lanes 4, 5 and 1 to lane 2). Once again, AMOT p130 (and breakdown products) co-immunoprecipitated with OSF-NEDD4L$_{C942A}$ (panel 1, lane 4), but not with a protein that also contained inactivating WW domain mutations (OSF-NEDD4L$_{\Delta WW/C942A}$ panel 1, lane 5). AMOT p130 co-precipitation levels were higher with OSF-NEDD4L$_{C942A}$ than with wild type OSF-NEDD4L, presumably owing to the higher cellular levels of both OSF-NEDD4L and AMOT p130 (compare lanes 2 and 4).

Three additional observations confirmed that the NEDD4L WW domains and AMOT p130 PPXY motifs are critical for the NEDD4L-AMOT p130 interaction. Firstly, mutating the three N-terminal AMOT p130 PPXY motifs strongly inhibited co-immunoprecipitation of HA-AMOT p130 with OSF-NEDD4L (*Figure 1—figure supplement 2*, panel 1, compare lanes 2 and 3). Secondly, endogenous AMOT p80, which lacks the N-terminal PPXY motifs, failed to co-immunoprecipitate with either OSF-NEDD4L or OSF-NEDD4L$_{C942A}$ (*Figure 1C*, panel 1, lanes 2 and 4), even though AMOT p80 was present in the 293T cell extracts (panel 3, lanes 1–5). Thirdly, we observed that exogenously expressed, epitope-tagged NEDD4L and AMOT p130 co-localized well in HeLa-M cells, and this co-localization required both the AMOT PPXY and NEDD4L WW motifs (data not shown), in good agreement with previous reports that motins co-localize with other NEDD4 family members (*Skouloudaki and Walz, 2012*; *Wang et al., 2012*). Taken together, these experiments demonstrate that NEDD4L interacts specifically with AMOT p130 in cells and that the interaction requires the WW domains of NEDD4L and the PPXY motifs of AMOT.

## AMOT binds directly to NEDD4L and to HIV-1 Gag

Pull-down experiments with purified recombinant proteins were performed to test whether AMOT p130 binds NEDD4L directly, and whether AMOT p130 can also bind HIV-1 Gag and thereby link NEDD4L to the viral Gag protein. Wild type and mutant NEDD4L proteins and an HIV-1 Gag protein that lacked the flexible p6 domain (Gag$_{\Delta p6}$) were expressed in *Escherichia coli* and purified to near homogeneity (*Figure 2A*, lanes 1 and 2, and *Figure 2B*, lane 1, respectively). OSF-AMOT p130 was overexpressed in SF-9 insect cells and substantially enriched by binding to STrEP-Tactin affinity resin (*Figure 2A*, lane 6 and *Figure 2B*, lane 7), although we were unable to purify the protein to homogeneity. As shown in *Figure 2A*, NEDD4L did not bind an immobilized control OSF-GFP protein, but bound with at least 1:1 stoichiometry to immobilized OSF-AMOT p130 (compare lanes 4 and 7). In contrast, binding of the NEDD4L$_{\Delta WW}$ mutant protein was nearly undetectable (*Figure 2A*, compare lanes 7 and 8, particularly in the western blot in panel 2). These experiments confirm that NEDD4L and AMOT p130 bind one another directly, with the WW domains of NEDD4L binding the PPXY motifs of AMOT p130.

Analogous pull-down experiments were performed to test whether AMOT binds HIV-1 Gag$_{\Delta p6}$. Removing the p6 region facilitates Gag protein expression and purification, and is not expected to interfere with any biologically relevant interactions because NEDD4L potently stimulates HIV-1

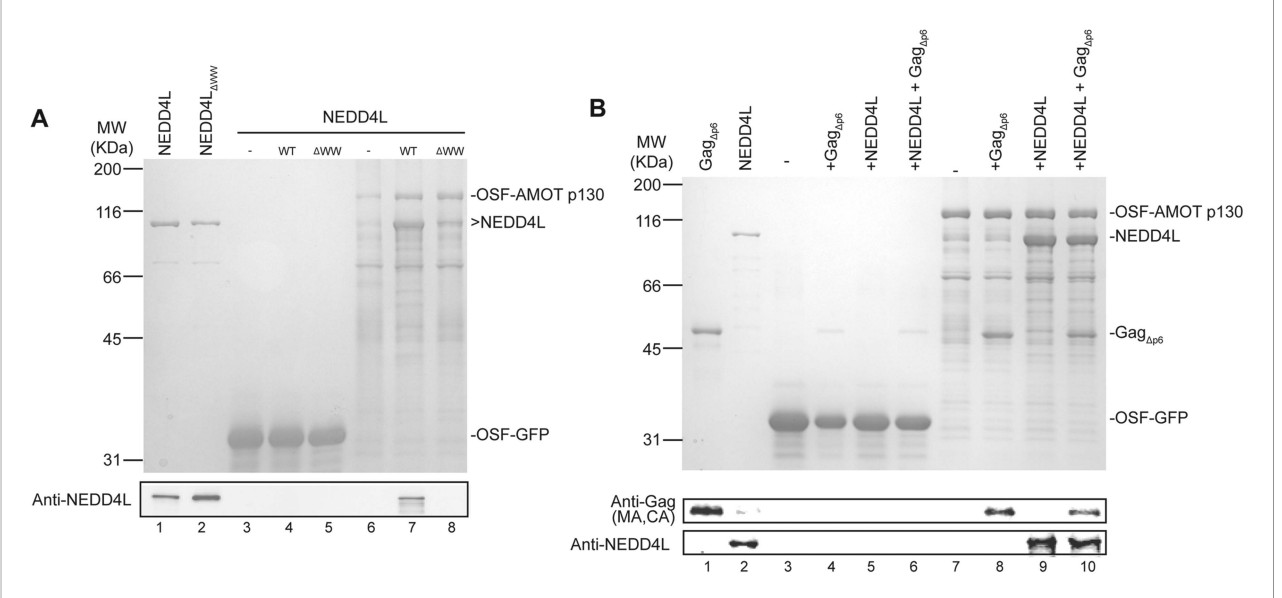

**Figure 2**. AMOT p130 binds directly and specifically to NEDD4L and HIV-1 Gag$_{\Delta p6}$. (**A**) OSF-AMOT p130 binds NEDD4L. Recombinant OSF-GFP (specificity control, lanes 3–5) or OSF-AMOT p130 (lanes 6–8) were expressed, captured on STrEP-Tactin affinity matrices, and incubated with either a buffer control (lanes 3 and 6) or buffers containing 0.5 μM wild type NEDD4L (lanes 4 and 7) or NEDD4L$_{\Delta WW}$ (lanes 5 and 8, inactivating point mutations in all four WW domains). Matrix-bound proteins were released by boiling in denaturing buffer and detected by SDS-PAGE with Coomassie blue staining (panel 1) or by western blotting (panel 2, anti-NEDD4L) to confirm the identities of the bound NEDD4L and help to distinguish background proteins from low-level binding in panel 1. Input levels of NEDD4L (lane 1, 1.5% of total) and NEDD4L$_{\Delta WW}$ (lane 2, 1.5% of total) are shown for reference. A representative image from three independent repetitions is shown. (**B**) OSF-AMOT p130 binds HIV-1 Gag$_{\Delta p6}$. Recombinant OSF-GFP (control, lanes 3–6) or OSF-AMOT p130 (lanes 7–10) were expressed, captured on STrEP-Tactin affinity matrices and incubated with a buffer control (lanes 3 and 7) or with buffers containing 1.0 μM HIV-1 Gag$_{\Delta p6}$ (lanes 3 and 8), 0.5 μM wild type NEDD4L (lanes 5 and 9) or both proteins (lanes 6 and 10). Matrix-bound proteins were released by boiling in denaturing buffer and detected by SDS-PAGE with Coomassie blue staining (panel 1) or by western blotting to confirm the identities of the bound Gag (panel 2, anti-MA and anti-CA) and NEDD4L (panel 3, anti-NEDD4L) proteins and help to distinguish background proteins from low-level binding in panel 1. Input Gag$_{\Delta p6}$ (lane 1, 2% of total) and NEDD4L (lane 2, 1.5% of total) are shown for reference. A representative image from three independent repetitions is shown.

The following figure supplement is available for figure 2:

**Figure supplement 1**. AMOT p130, AMOTL1 and AMOTL2 bind directly and specifically to NEDD4L and HIV-1 Gag$_{\Delta p6}$.

Gag$_{\Delta p6}$ release from cells (*Chung et al., 2008*; *Usami et al., 2008*). As shown in *Figure 2B*, HIV-1 Gag$_{\Delta p6}$ bound immobilized OSF-AMOT p130, but did not bind an immobilized OSF-GFP control (compare lanes 4 and 8). This interaction did not appear to be mediated by nucleic acids because the affinity-purified OSF-AMOT p130 protein was pre-washed with high salt solution to remove any bound nucleic acid (see 'Materials and methods'), and because the interaction was maintained, albeit at slightly lower levels, even when immobilized OSF-AMOT p130 was incubated with high concentrations of RNase that were sufficient to eliminate other RNA-mediated Gag-protein interactions (not shown). Gag$_{\Delta p6}$ and NEDD4L appeared to bind OSF-AMOT p130 independently because their binding levels did not change significantly when all three proteins were incubated together (compare lanes 8 and 9 to lane 10).

AMOT is the founding member of the human motin family, which also includes Angiomotin-like protein 1 (AMOTL1) and Angiomotin-like protein 2 (AMOTL2) (*Moleirinho et al., 2014*). AMOTL1 and AMOTL2 share 52% and 45% pair-wise sequence identity with AMOT p130, and both contain the N-terminal extension and PPXY motifs that are present in AMOT p130 but absent in AMOT p80 (*Figure 1A*). Pull-down experiments demonstrated that AMOTL1 and AMOTL2 also bind both NEDD4L and HIV-1 Gag$_{\Delta p6}$ (*Figure 2—figure supplement 1*). Thus, AMOT p130 binds directly and specifically to both NEDD4L and HIV-1 Gag, and these binding activities are shared by the other two human motins.

# AMOT p130 overexpression enhances NEDD4L stimulation of HIV-1 budding

Co-expression experiments were performed to test whether AMOT and NEDD4L can cooperate in stimulating HIV-1 release (*Figure 3*). These experiments employed the crippled HIV-1$_{\Delta PTAP, \Delta YP}$ viral expression construct, which lacks functional PTAP and YPX$_n$L late domains and is therefore particularly responsive to cellular NEDD4L activity (*Chung et al., 2008*; *Usami et al., 2008*). As expected, HIV-1$_{\Delta PTAP, \Delta YP}$ was poorly released from control 293T cells, as measured by western blot that detected virion-associated CA and MA proteins (*Figure 3*, upper right panel, lane 1), and low viral titers (lower right panel, lane 1). Overexpression of HA-AMOT p130 increased the levels of the full-length protein (and presumptive breakdown products were also evident, left panel 1, compare lanes 1 and 2). AMOT p130 overexpression modestly increased HIV-1$_{\Delta PTAP.\Delta YP}$ release and infectivity (right panels compare lanes 1 and 2, 1.7-fold increases in both release and infectivity), without significantly altering the intracellular levels of Gag (left, panel 4, compare lanes 1 and 2) or a control cellular protein (GAPDH, left panel 3, compare lanes 1 and 2). Thus, AMOT p130 overexpression stimulates HIV-1$_{\Delta PTAP, \Delta YP}$ release and infectivity, but the effect is modest.

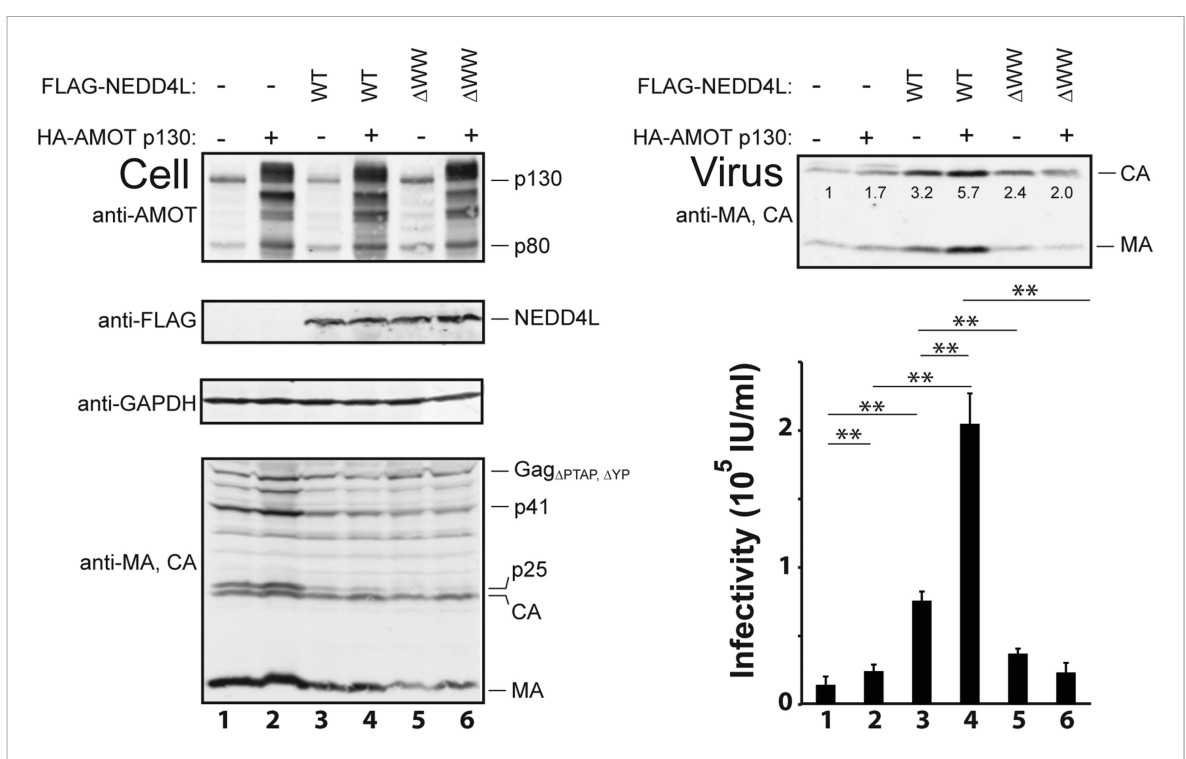

**Figure 3**. AMOT p130 stimulates NEDD4L-dependent release of HIV-1$_{\Delta PTAP, \Delta YP}$. Left panels are western blots showing 293T cellular levels of endogenous AMOT and exogenous HA-AMOT p130 proteins (panel 1, anti-AMOT), exogenous FLAG-NEDD4L (panel 2, anti-FLAG), endogenous GAPDH (panel 3, anti-GAPDH, loading control) and HIV-1 Gag$_{\Delta PTAP, \Delta YP}$ proteins (panel 4, anti-MA and anti-CA). Cells were co-transfected with expression vectors for HIV-1$_{\Delta PTAP, \Delta YP}$ (lanes 1–6), FLAG-NEDD4L proteins (lanes 3–6, with wild type (WT) FLAG-NEDD4L in lanes 3 and 4 and FLAG-NEDD4L$_{\Delta WW}$, in lanes 5 and 6), and wild type HA-AMOT p130 (lanes 2, 4 and 6) or an empty vector control (lanes 1, 3, and 5). Right panels show corresponding levels of extracellular, virion-associated CA$^{Gag}$ and MA$^{Gag}$ proteins (panel 1, anti-MA and anti-CA) and viral titers (panel 2, IU denotes 'infectious units'). Here and in subsequent figures: (1) error bars denote standard deviations between independent repetitions of the experiment, n = 5 in this case, and (2) numbers within the blots show integrated intensities of the CA band intensities (relative to the value in the control experiment, set to 1.0). Here and throughout significance is denoted by: NS, not significant (p > 0.05); *0.05 > p > 0.01; **p < 0.01.

The following figure supplements are available for figure 3:

**Figure supplement 1**. Dose-dependent AMOT p130 stimulation of NEDD4L-dependent release of HIV-1$_{\Delta PTAP, \Delta YP}$.

**Figure supplement 2**. The AMOT p130 isoform specifically stimulates NEDD4L-dependent release of HIV-1$_{\Delta PTAP, \Delta YP}$.

As seen previously, FLAG-NEDD4L overexpression also stimulated HIV-1$_{\Delta PTAP, \Delta YP}$ release (*Figure 3*, right panel 1, compare lanes 1 and 3, 3.2-fold stimulation) and infectivity (right panel 2, compare lanes 1 and 3, 5.3-fold stimulation), and also largely rescued the HIV-1$_{\Delta PTAP, \Delta YP}$ Gag processing defects (left panel 4, compare lane 3 to lanes 1 and 2, and note the reduction in p41 and p25 Gag processing intermediates) (*Chung et al., 2008*; *Usami et al., 2008*). Co-overexpression of both HA-AMOT p130 and FLAG-NEDD4L further increased HIV-1$_{\Delta PTAP, \Delta YP}$ release (right panel 1, lane 4, sixfold increase over basal levels) and infectivity (right panel 2, lane 4, 14-fold increase over basal levels). As shown in *Figure 3—figure supplement 1*, the degree of hyper-stimulation correlated positively with AMOT p130 expression levels, until an inhibitory level was reached. In contrast to wild type NEDD4L, the NEDD4L$_{\Delta WW}$ mutant stimulated HIV-1$_{\Delta PTAP, \Delta YP}$ release and infectivity only very modestly (*Figure 3*, right panels, compare lanes 5 and 3) and did not synergize with AMOT p130 (right panels, compare lanes 6 and 5). Similarly, the shorter AMOT p80 isoform, which lacks the PPXY motifs found in the N-terminal domain of AMOT p130, failed to synergize with NEDD4L in stimulating virus release and infectivity, and instead modestly *reduced* NEDD4L stimulation, probably by dominantly inhibiting endogenous AMOT p130 activity (*Figure 3—figure supplement 2*, compare lanes 4–6). These data indicate that AMOT p130 and NEDD4L cooperate in stimulating HIV-1$_{\Delta PTAP, \Delta YP}$ release and infectivity.

## AMOT p130 is required for NEDD4L stimulation of HIV-1 budding

siRNA depletion experiments were performed to determine whether AMOT was also *necessary* for NEDD4L stimulation of HIV-1$_{\Delta PTAP, \Delta YP}$ release and infectivity. We used an siRNA that targeted both AMOT isoforms and reduced cellular AMOT p130 and p80 levels by at least fivefold (*Figure 4*, left panel 1, compare lanes 1 and 2 and lanes 3 and 4). AMOT depletion reduced the already low levels of virus release and infectivity even further, to near background levels (*Figure 4*, right panels, compare lanes 1 and 2, 7.5-fold infectivity reduction). Cellular levels of Gag and a GAPDH control were unaffected by this treatment (left panels 3 and 4, compare lanes 1 and 2), although AMOT depletion reduced cellular protein levels slightly in some repetitions (not shown).

AMOT depletion blocked the ability of NEDD4L to stimulate HIV-1$_{\Delta PTAP, \Delta YP}$ release and infectivity (*Figure 4* and *Figure 4—figure supplements 1, 2*). In the control case, NEDD4L overexpression stimulated HIV-1$_{\Delta PTAP, \Delta YP}$ release (*Figure 4*, right panels, compare lanes 1 and 3, 4.1-fold infectivity increase). However, virus release was reduced to the levels seen in untreated control cells when endogenous AMOT p80 and p130 proteins were simultaneously depleted (compare lanes 2–4). Similar results were observed for a second siRNA that targeted both AMOT isoforms (not shown). Thus, AMOT is required for NEDD4L stimulation of HIV-1$_{\Delta PTAP, \Delta YP}$ release.

Rescue experiments with exogenous siRNA-resistant AMOT expression constructs were performed to confirm the specificity of the AMOT depletion and to test the requirements for AMOT function. Re-expression of AMOT p130 completely rescued the block to virus release and infectivity imposed by AMOT depletion (*Figure 4*, right panels, compare lanes 5 and 4), restoring viral infectivity to a level that was actually slightly higher than the control (compare lanes 3 and 5). The degree of rescue generally correlated positively with AMOT p130 re-expression levels, and peaked at levels that approximated those of the native endogenous protein (*Figure 4—figure supplement 1*, lane 8). In contrast, an AMOT p130 protein that lacked the three N-terminal PPXY motifs failed to rescue NEDD4L-dependent HIV-1$_{\Delta PTAP, \Delta YP}$ release (*Figure 4*, AMOT p130$_{\Delta PPXY}$, right panels, compare lanes 6 and 5). Similarly, re-expression of an siRNA-resistant AMOT p80 construct also failed to rescue virus release and infectivity (*Figure 4—figure supplement 2*, right panels, compare lanes 4 and 5), unless exogenous AMOT p130 was present (*Figure 4—figure supplement 2*, right panels, lane 7). Hence, NEDD4L stimulation of HIV-1$_{\Delta PTAP, \Delta YP}$ release and infectivity requires the presence of an AMOT p130 protein that can bind NEDD4L, but is not significantly affected by AMOT proteins that cannot bind NEDD4L, including AMOT p80.

## AMOT p130 is required for efficient release of wild type HIV-1

siRNA depletion and rescue experiments were also performed to test whether AMOT p130 is required for release and infectivity of *wild type* HIV-1. In control experiments, depletion of either of the well-characterized HIV-1 budding factors, ALIX and TSG101, reduced virus release and infectivity, although the effects were much greater for TSG101 depletion, as reported previously (*Figure 5*, right panels, compare lanes 2 and 3 to lane 1) (*Fujii et al., 2009*). AMOT depletion also significantly

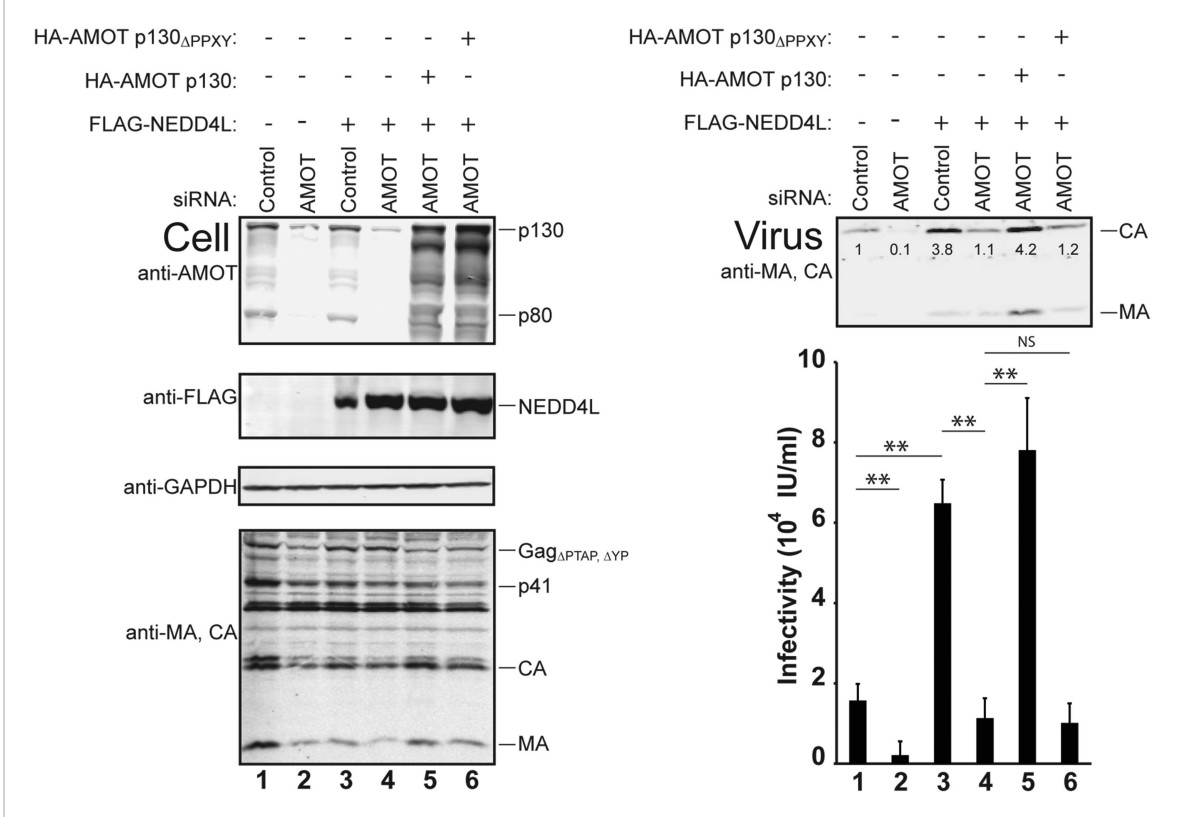

**Figure 4**. AMOT p130 is required for NEDD4L-stimulated release of HIV-1$_{\Delta PTAP, \Delta YP}$. Left panels are western blots showing 293T cellular levels of endogenous AMOT and exogenous HA-AMOT p130 proteins (panel 1, anti-AMOT), exogenous FLAG-NEDD4L (panel 2, anti-FLAG), endogenous GAPDH (panel 3, anti-GAPDH, loading control) and HIV-1 Gag$_{\Delta PTAP, \Delta YP}$ proteins (panel 4, anti-MA and anti-CA). Cells were co-transfected with a control siRNA (lanes 1 and 3) or with an siRNA that depleted both AMOT p130 and p80 (lanes 2 and 4–6), with expression vectors for HIV-1$_{\Delta PTAP, \Delta YP}$ (lanes 1–6), and with FLAG-NEDD4L (lanes 3–6) or an empty vector control (lanes 1 and 2), and with siRNA-resistant expression constructs for HA-AMOT p130 proteins (with wild type HA-AMOT p130 in lane 5 and an HA-AMOT p130$_{\Delta PPXY}$ protein carrying inactivating point mutations in the three PPXY motifs in lane 6). Right panels show corresponding levels of extracellular virion-associated Gag proteins (upper panel, western blot, anti-MA and anti-CA) and viral titers (lower panel), n = 6.

The following figure supplements are available for figure 4:

**Figure supplement 1**. Dose-dependent AMOT p130 rescue of HIV-1 release from cells depleted of endogenous AMOT.

**Figure supplement 2**. The AMOT p130 isoform specifically stimulates NEDD4L-dependent release of HIV-1$_{\Delta PTAP, \Delta YP}$.

reduced both HIV-1 release and infectivity (*Figure 5*, right panels, compare lane 4 to lane 1, eightfold reduction in infectivity and 2.5-fold reduction in release). The magnitudes of these effects were intermediate between those seen for depletion of TSG101 (20-fold reduction in infectivity and threefold reduction in virion release) and ALIX (1.5-fold reduction in infectivity and insignificant reduction in virion release). Importantly, the detrimental effects of AMOT p130 depletion were almost completely rescued by re-expression of wild type AMOT p130 (compare lanes 1, 4 and 5) but not by the mutant AMOT p130$_{\Delta PPXY}$ protein (lane 6). These experiments demonstrate that AMOT p130 is required for efficient release of wild type HIV-1 from 293T cells and mutations that abolish NEDD4L binding block the ability of AMOT p130 to function in virus release.

## AMOT stimulates HIV-1 release from HeLa cells

AMOT mRNA is expressed in 293T cells, as well as in white blood cells and lymphoid tissues that are the natural hosts for HIV-1 infection (*Moleirinho et al., 2014*). In contrast, HeLa cells reportedly express little or no AMOT mRNA, although they do express AMOTL2 mRNA (*Moleirinho et al., 2014*).

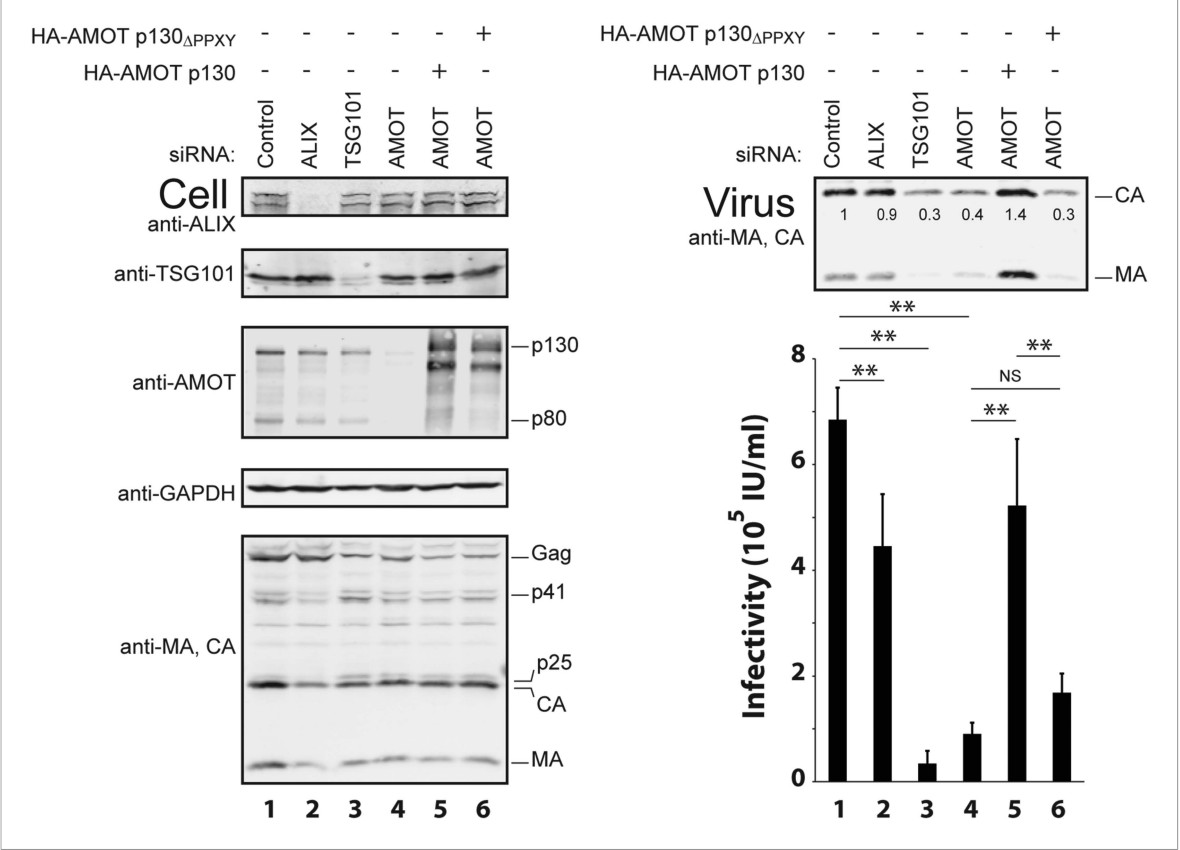

**Figure 5**. AMOT p130 is required for efficient release of wild type HIV-1. Left panels are western blots showing 293T cellular levels of endogenous ALIX (panel 1, anti-ALIX), TSG101 (panel 2, anti-TSG101), AMOT (panel 3, anti-AMOT), GAPDH (panel 4, anti-GAPDH, loading control) and levels of HIV-1 Gag proteins (panel 5, anti-MA and anti-CA). Cells were co-transfected with a control siRNA (lane 1) or with an siRNA that depleted ALIX (lane 2), TSG101 (lane 3) or AMOT p130 and p80 (lanes 4–6), with expression vectors for wild type HIV-1 (lanes 1–6) and with an siRNA-resistant expression constructs for HA-AMOT p130 proteins (with wild type HA-AMOT p130 in lane 5 and an HA-AMOT p130$_{\Delta PPXY}$ protein with inactivating point mutations in all three PPXY motifs in lane 6). Right panels show corresponding levels of extracellular virion-associated Gag proteins (upper panel, western blot, anti-MA and anti-CA) and viral titers (lower panel), n = 4.

We confirmed that endogenous AMOT protein levels are at least 10-fold lower in HeLa cells than in 293T cells (*Figure 6—figure supplement 1*). This observation begs the question of how HIV-1 can be released from a cell type that lacks AMOT, and this issue is particularly relevant because HeLa cells are commonly used to study HIV-1 assembly. We therefore tested the effects of AMOT p130 expression on HIV-1 release and infectivity in HeLa cells. As shown in *Figure 6A*, AMOT p130 expression increased HIV-1 release from HeLa cells in a dose-dependent fashion, up to 15-fold at the highest levels of AMOT tested (right panel 1, compare lanes 1 and 5). Viral titers similarly increased with AMOT expression in a dose-dependent fashion, although the overall effects were less dramatic (right panel 2, compare lanes 1–5, threefold infectivity increase). Thus, HIV-1 release from wild type HeLa cells is suboptimal, and can be enhanced by expression of AMOT p130.

## AMOT also stimulates HIV-1 release from Jurkat T cells

T cells are natural hosts for HIV-1 infection, and AMOT mRNA is present in these cells, including Jurkat T cells (*Yeung et al., 2009*; *Sheynkman et al., 2013*). To determine whether AMOT p130 can function in virus release from Jurkat T cells, we tested whether expressing exogenous AMOT p130 expression stimulated HIV-1 release and infectivity. As shown in *Figure 6B*, AMOT p130 expression increased HIV-1 release from Jurkat T cells in a dose-dependent fashion, with up to 16-fold stimulation observed at the highest levels of AMOT p130 tested (panel 1, compare lanes 1–3). Viral titers similarly increased in a dose-dependent fashion (panel 2, compare lanes 1–3, sevenfold infectivity increase). Western

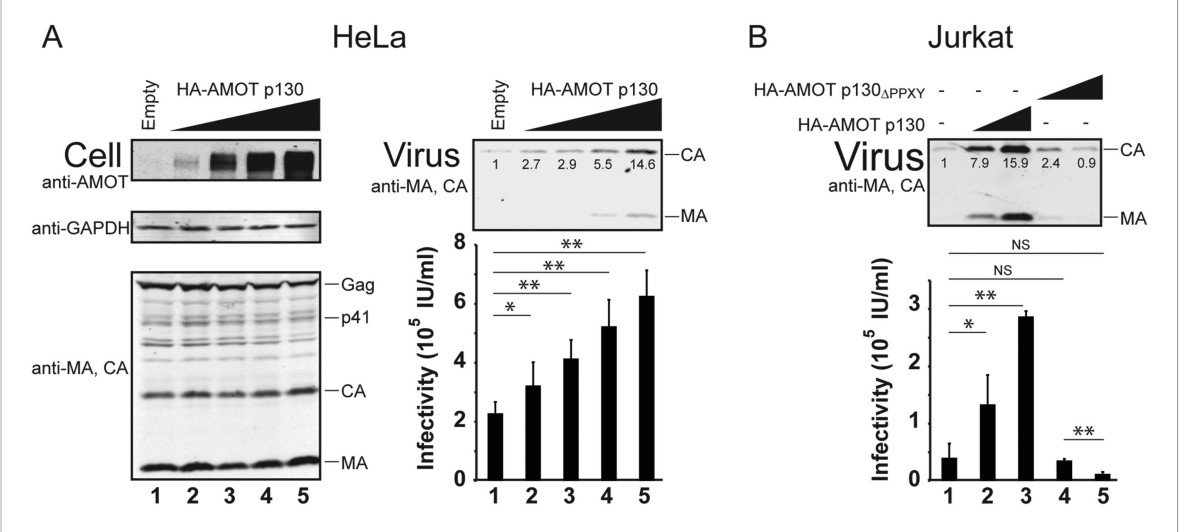

**Figure 6**. AMOT p130 stimulates release of HIV-1 from HeLa and Jurkat T cells. (**A**) Left panels are western blots showing HeLa cellular levels of endogenous AMOT p130 and exogenous HA-AMOT p130 (panel 1, anti-AMOT), endogenous GAPDH (panel 2, anti-GAPDH, loading control) and HIV-1 Gag proteins (panel 3, anti-MA and anti-CA). HeLa cells were co-transfected with expression vectors for HIV-1 (lanes 1–5), and with increasing concentrations of an expression construct for HA-AMOT p130 (0–4 µg DNA in lanes 1–5). Right panels show corresponding levels of extracellular virion-associated Gag proteins (upper panel, western blot, anti-MA and anti-CA) and viral titers (lower panel), n = 4. (**B**) Levels of extracellular virion-associated Gag proteins (upper panel, western blot, anti-MA and anti-CA) and viral titers (lower panel). Jurkat T cells were co-transfected with expression vectors for HIV-1 (lanes 1–5), together with a control vector (lane 1) or expression vectors for wild type HA-AMOT p130 (lanes 2 and 3, 5 and 10 µg DNA respectively) or an HA-AMOT p130$_{\Delta PPXY}$ protein with inactivating point mutations in the three PPXY motifs (lanes 4 and 5, 5 and 10 µg DNA respectively), n = 3.

The following figure supplement is available for figure 6:

**Figure supplement 1**. HeLa cells express little or no AMOT.

blots confirmed that virus expression levels were the same in all cases (not shown), but AMOT p130 expression levels were below our detection limits in this cell type. We therefore also tested the effects of overexpressing the mutant AMOT p130$_{\Delta PPXY}$ protein. Unlike wild type AMOT p130, AMOT p130$_{\Delta PPXY}$ failed to stimulate virus release, and actually reproducibly *decreased* viral titers as compared to the untreated control (compare lanes 4 and 5 to lane 1). We speculate that this effect again reflects dominant inhibition of endogenous AMOT p130. Regardless, our data indicate that AMOT p130 also stimulates virus release and infectivity from T cells, which are natural targets of HIV-1 infection. Consistent with this conclusion, a global shRNA screen for host cofactors revealed that Jurkat T cells transduced with an shRNA that targeted AMOT were protected against HIV cytotoxicity (*Yeung et al., 2009*). This observation implies that AMOT contributes to HIV-1 replication in cultured Jurkat T cells, although the role of AMOT in HIV-1 replication was not characterized further in that study. (*Yeung et al., 2009*).

## Other human motins can also stimulate HIV-1 release

siRNA depletion/rescue experiments were performed to test whether other motin family members can substitute for AMOT p130 in supporting release of wild type HIV-1 (*Figure 7*). In control experiments, HIV-1 release and infectivity were again decreased by depletion of endogenous AMOT from 293T cells (right panels, compare lanes 1 and 2, fivefold infectivity reduction). As expected, these effects were largely rescued by expression of exogenous siRNA-resistant AMOT p130 (right panels, compare lanes 2 and 3). Expression of either AMOTL1 or AMOTL2 also significantly rescued HIV-1 release and infectivity (right panels, compare lanes 4 and 5 to lanes 2 and 3). Hence, both AMOT-like proteins can facilitate HIV-1 release in the absence of endogenous AMOT p130. Neither AMOTL1 nor AMOTL2 was quite as effective as AMOT p130 in rescuing HIV-1 infectivity in this assay, possibly because these proteins are intrinsically less active than AMOT p130 and/or because they were expressed at lower levels (left panel 2, compare lanes 3–5).

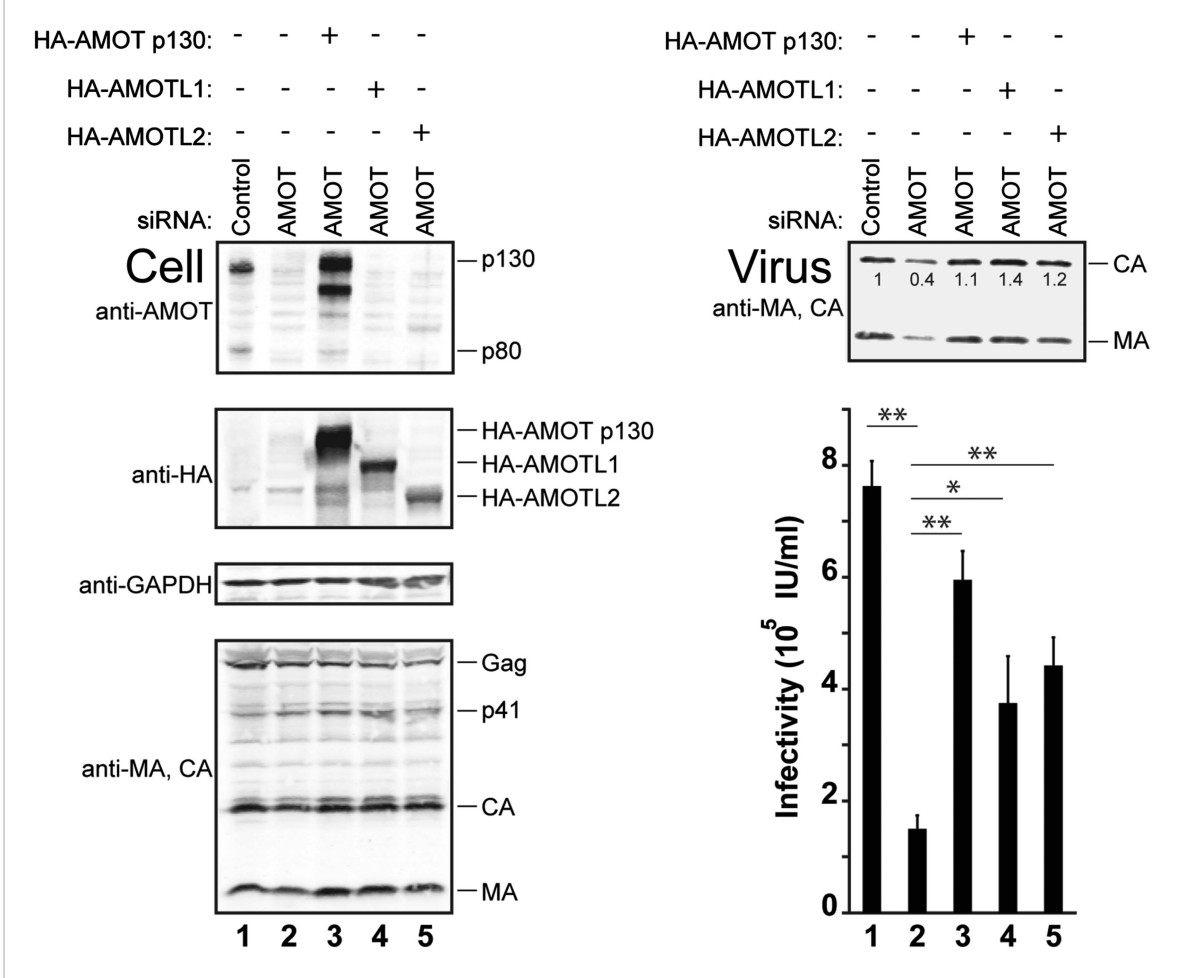

**Figure 7**. AMOTL1 and AMOTL2 can substitute for AMOT p130 in HIV-1 release. Left panels are western blots showing 293T cellular levels of endogenous AMOT and exogenous HA-AMOT p130 proteins (panel 1, anti-AMOT), exogenous HA-AMOT p130, HA-AMOTL1 or HA-AMOTL2 proteins (panel 2, anti-HA), endogenous GAPDH (panel 3, anti-GAPDH, loading control) and HIV-1 Gag proteins (panel 4, anti-MA and anti-CA). Cells were co-transfected with a control siRNA (lane 1) or with an siRNA that depleted AMOT p130 and AMOT p80 (lanes 2–5), and with expression vectors for HIV-1 (lanes 1–5), with siRNA-resistant expression constructs for AMOT p130 (lane 3), AMOTL1 (lane 4) or AMOTL2 (lane 5). Right panels show corresponding levels of extracellular virion-associated Gag proteins (upper panel, western blot, anti-MA and anti-CA) and viral titers (lower panel), n = 4.

The following figure supplement is available for figure 7:

**Figure supplement 1**. AMOTL2 can stimulate NEDD4L-dependent release of HIV-1$_{\Delta PTAP, \Delta YP}$ and rescue HIV-1$_{\Delta PTAP, \Delta YP}$ release from cells depleted of endogenous AMOT.

We also tested whether the other AMOT family members could cooperate with NEDD4L in stimulating release of the crippled HIV-1$_{\Delta PTAP, \Delta YP}$ virus (*Figure 7—figure supplement 1A*) and whether AMOTL1 and AMOTL2 could rescue the block to NEDD4L stimulation of HIV-1$_{\Delta PTAP, \Delta YP}$ release imposed by depletion of endogenous AMOT (*Figure 7—figure supplement 1B*). AMOTL2 strongly stimulated/rescued HIV-1$_{\Delta PTAP, \Delta YP}$ release in both of these assays, albeit slightly less efficiently than AMOT p130 (*Figure 7—figure supplement 1A*, compare lane 8 to lanes 5 and 6 and *Figure 7—figure supplement 1B*, compare lane 7 to lanes 4 and 5). In contrast, AMOTL1 exhibited little or no activity in either assay (*Figure 7—figure supplement 1A*, compare lane 7 to lanes 5 and 6 and *Figure 7—figure supplement 1B*, compare lane 6 to lanes 4 and 5). Thus, AMOTL2 could substitute for AMOT p130 in three different virus release assays, whereas AMOTL1 was slightly less active than AMOT in the release of wild type HIV-1, and failed to stimulate release of the 'crippled' HIV-1$_{\Delta PTAP, \Delta YP}$ construct.

## AMOT depletion inhibits an early stage of HIV-1 budding

Scanning and transmission electron microscopy (SEM and TEM, respectively) were used to visualize the stage at which AMOT depletion inhibited HIV-1 assembly or release (*Figures 8, 9*). SEM imaging was initially performed because this method can survey the entire cell surface and thereby identify global changes in virion assembly and budding profiles. SEM analyses were performed on 293T cells that expressed HIV-1 and were treated with either: (1) a control siRNA, (2) an siRNA that depleted TSG101 (positive control), or (3) an siRNA that depleted AMOT. As expected, viruses budding from control siRNA-treated cells were detectable but rare (e.g., *Figure 8A–C*). In contrast, large numbers of assembling virions were visible on the surfaces of cells that lacked TSG101 (*Figure 8D–F*) or that lacked AMOT (*Figure 8G–I*). We also observed that the arrested virions tended to cluster in the center of cells lacking AMOT, whereas they were more evenly distributed across the surfaces of cells lacking TSG101. The dramatic increase in the numbers of arrested budding virions implies that AMOT, like TSG101, must function following the initial stages of Gag assembly, but prior to viral membrane fission.

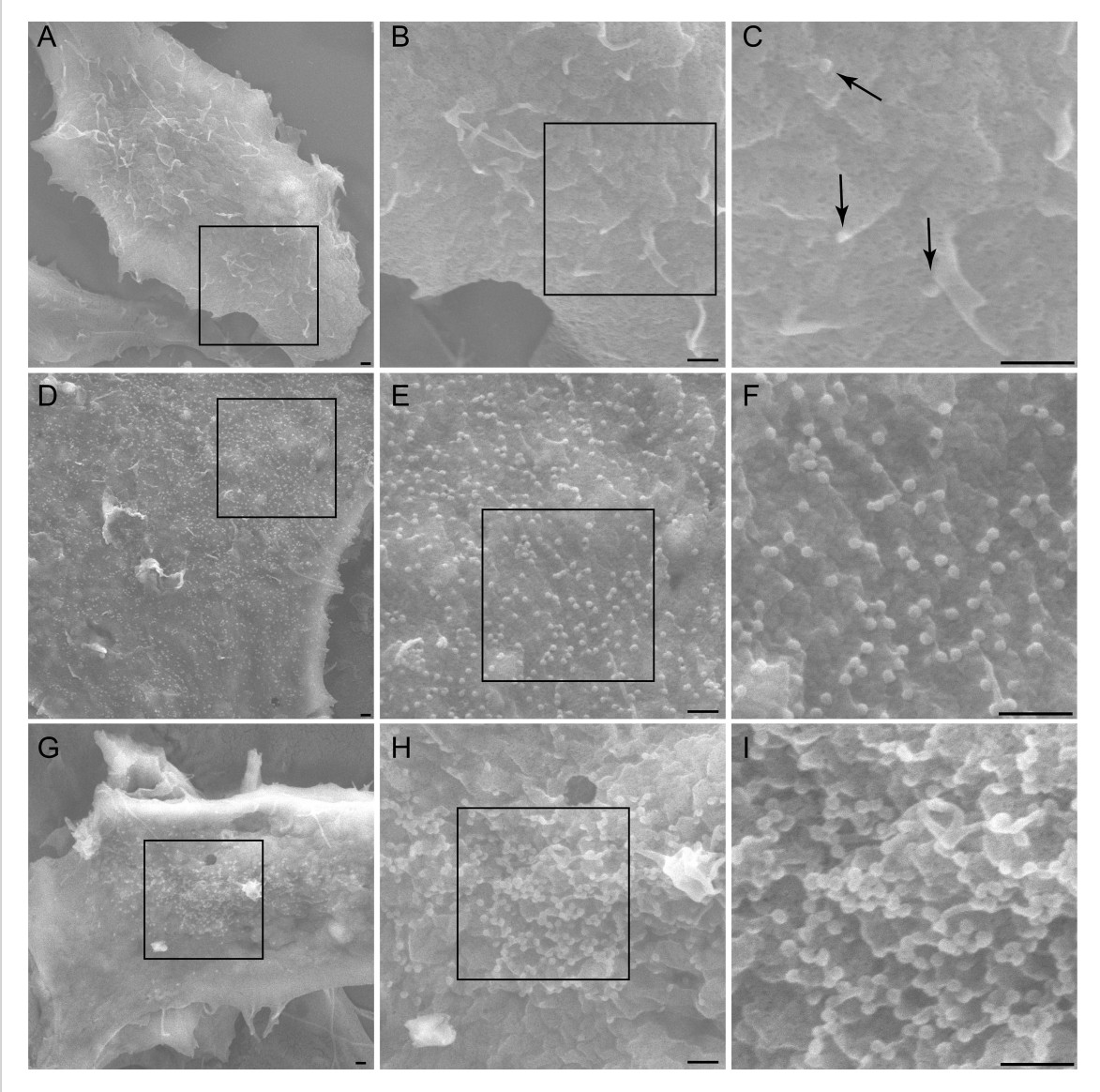

**Figure 8**. AMOT p130 is required for HIV-1 assembly/budding. Scanning electron microscopic (SEM) images of HIV virions budding from 293T cells depleted with a control siRNA (**A–C**), depleted of TSG101 (**D–F**) or depleted of AMOT (**G–I**). Successive images across each row show expansions of the adjacent boxed regions, and scale bars are 500 nm in all cases. Arrows in panel (**C**) highlight budding wild type HIV-1 virions.

TEM experiments were performed to characterize how virus assembly/budding arrested in the absence of AMOT. These experiments again visualized HIV-1 budding from 293T cells treated with a control siRNA, an siRNA that depleted TSG101 (positive control), an siRNA that depleted AMOT, or an siRNA that depleted endogenous AMOT together with an siRNA-resistant construct that re-expressed wild type AMOT p130 (rescue control). Culture supernatants were removed from the virus producing cells and the adherent cells were fixed while still attached to the culture dish in order to maximize the numbers of cell-associated virions. Fixed cells were harvested, embedded in resin, thin sectioned, stained, and visualized by TEM. Cell-associated virions observed in these experiments were scored as being either: (1) mature (discernable conical or condensed cores), (2) immature (discernable immature Gag shells), or (3) budding (contiguous cellular and viral membranes).

HIV-1 virions associated with wild type cells were usually mature (76 ± 5%, see *Figure 9C*), but immature and budding virions were also observed (9 ± 3% and 15 ± 5%, respectively). As expected, TSG101 depletion induced a strong virus budding defect, with the majority of detected virions arrested at the budding stage (67 ± 7%, shown in *Figure 9A* and quantified in *Figure 9C*). In addition, the ratio of mature (10 ± 6%) to immature (23 ± 2%) virions (1:2.3) was much lower than in the control case (1:0.1), consistent with previous reports that TSG101 is also required for efficient Gag processing and maturation (*Göttlinger et al., 1991*; *Garrus et al., 2001*; *Martin-Serrano et al., 2003*).

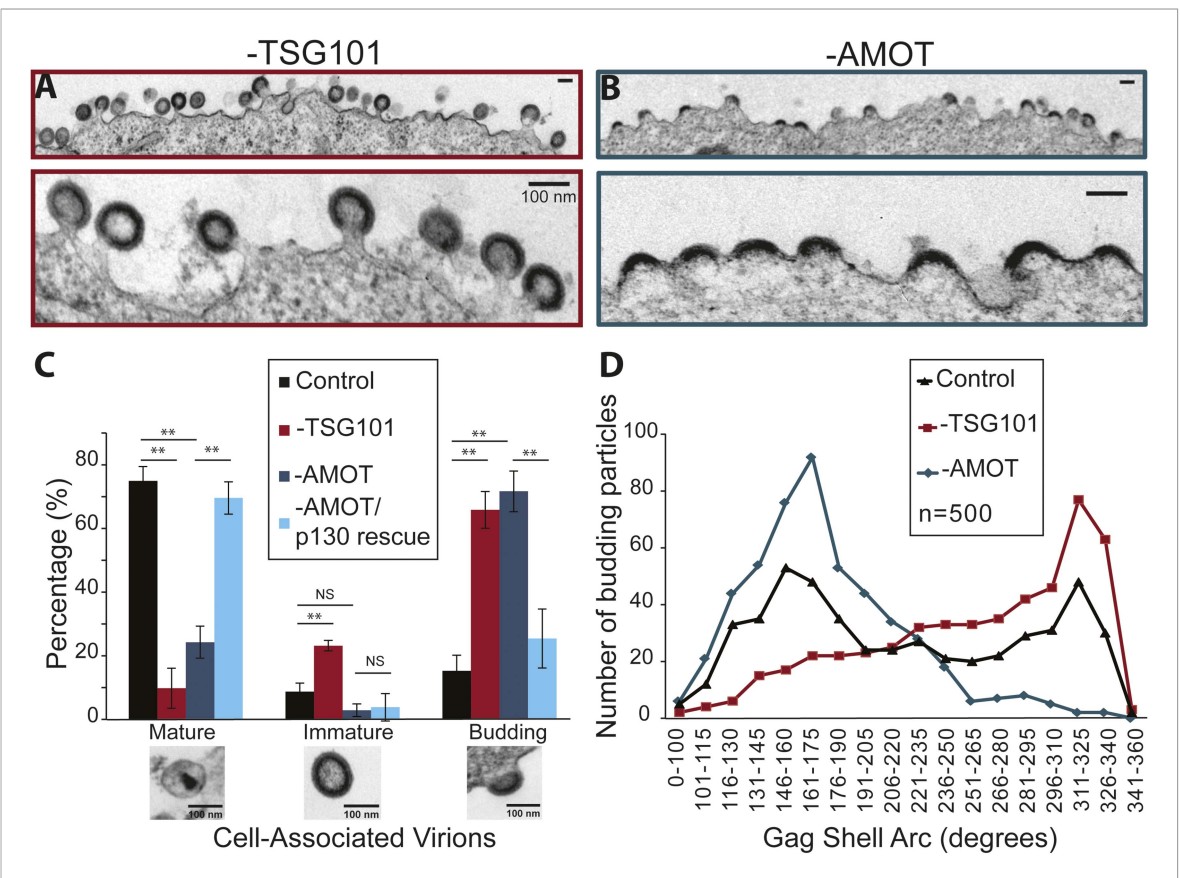

**Figure 9**. AMOT p130 is required for an early stage of HIV-1 assembly/budding. (**A**) Transmission electron microscopic (TEM) images of HIV virions budding from 293T cells depleted of TSG101. Panel 1: wide view. Panel 2: expanded view. Scale bars here and in part B are 100 nm. (**B**) TEM images of HIV virions budding from 293T cells depleted of AMOT p130 and p80. Panel 1: wide view. Panel 2: expanded view. (**C**) Quantification of the release and maturation status of HIV-1 virions associated with cells that were treated with a control siRNA (black bars), depleted of TSG101 (red), depleted of AMOT (teal), or depleted of AMOT and re-expressing AMOT p130 from an siRNA-resistant construct (light blue). This experiment was repeated twice with similar results, and error bars were derived by quantifying three separate sets of >100 virions each from one of the experiments. (**D**) Quantification of the completeness of virions budding from cells treated with a control siRNA (black triangles), depleted of TSG101 (red squares), or depleted of AMOT (teal diamonds). The extent of the Gag shell arc (in degrees) was measured from TEM images of 500 budding virions of each type, the measurements were binned into the intervals shown below the x-axis, and the virion numbers in each bin are shown in the plot.

AMOT depletion also inhibited virus release, as reflected in a large increase in the percentage of virions detected at the budding stage (72 ± 6%, shown in *Figure 9B* and quantified in *Figure 9C*). In this case, however, the virions that were released exhibited a normal ratio of mature:immature virions (25 ± 5% vs 3 ± 2%; 1:0.1). Importantly, the budding defect seen in cells lacking endogenous AMOT was almost completely rescued by re-expression of the siRNA-resistant AMOT p130 (*Figure 9C*), so that the majority of virions were now mature (70 ± 5% vs 76 ± 5% in the control), and budding virion levels were nearly normal (26 ± 9% vs 15 ± 5% in the control). Thus, AMOT depletion blocks virus release at the assembly/budding stage, but does not inhibit Gag processing or virion maturation.

It was also striking that AMOT depletion caused virion assembly/budding to arrest at an earlier stage than TSG101 depletion. This difference can be seen in the EM images shown in *Figure 9A,B*, where most of the virions in TSG101-depleted cells exhibit the characteristic 'lollipop' morphology associated with a late assembly/membrane fission defect (*Figure 9A*), whereas most of the virions budding from AMOT-depleted cells exhibit a 'half-moon' morphology in which the nascent viral membrane and Gag shell are incomplete (*Figure 9B*). This observation was quantified by imaging 500 budding HIV-1 virions from: (1) wild type cells (control), (2) cells depleted of TSG101, and (3) cells depleted of AMOT. The arc formed by the Gag shell was measured manually for each budding virion, and the 500 measurements for each condition were then binned in 15° intervals and displayed graphically (*Figure 9D*).

The Gag arc distribution of control virions budding from wild type cells exhibited two similarly sized peaks; one in which the Gag shells covered an arc of ~155°, and another in which the Gag shells covered an arc of ~320°. These measurements indicate that virion assembly intermediates (or possibly dead-end products) appreciably accumulate at two different stages, suggesting that either stage can be rate limiting. In cells lacking TSG101, budding-arrested virions were much more prevalent and they typically arrested at very late stages of assembly, with the maximal population exhibiting a Gag shell arc of ~320°. In cells lacking AMOT, arrested virions were again prevalent but they arrested at a much earlier stage, with the maximal population exhibiting a Gag shell arc of ~165°. The dramatic increases in numbers of budding virions induced by depletion of TSG101 or AMOT can explain how the assembly phenotypes could distribute as single peaks in these cases, whereas virions budding from wild type cells (which were much rarer) distributed into two peaks. Our experiments reveal that HIV budding arrests at an earlier stage in the absence of AMOT than in the absence of TSG101, implying that AMOT acts at an earlier stage of virion morphogenesis.

## Discussion

We have identified Angiomotin (AMOT) as a host protein required for efficient HIV-1 release. In the absence of AMOT, most virions arrested at a 'half-moon' stage in which a hemisphere of assembled Gag molecules distended the membrane, but did not form a spherical particle (*Figure 9*). The AMOT depletion phenotype was distinct from the defect induced by the absence of ESCRT factors, where virus budding arrested at a 'lollipop' stage in which nearly complete immature virions remained tethered to the host cell via thin membrane stalks. Thus, cellular factors facilitate (at least) two distinct steps in virion morphogenesis, with AMOT required to complete assembly/envelopment and the ESCRT machinery required for membrane fission. HIV-1 virions budding from control cells accumulated at both stages, suggesting that both steps can be kinetic bottlenecks (see *Figure 9D* and reference *Ku et al., 2013*).

The Gag lattice organizes the virion and undoubtedly helps to drive membrane envelopment because pure recombinant Gag proteins can form spherical particles in vitro (*Campbell et al., 2001*). In cells, however, Gag appears to require additional activities to bend the membrane, extrude the particle, and sever the bud neck. The well-studied process of endocytic vesicle formation provides precedence for these additional requirements because clathrin alone can form spherical assemblies but nevertheless requires assistance from BAR domain-containing proteins to help bend membranes, and assistance from actin and dynamin to release the vesicle (*Weinberg and Drubin, 2012*). It is intriguing that AMOT p130 also contains a candidate BAR-like domain (*Heller et al., 2010*) and can associate with F-actin (Ernkvist et al., 2006, *2008*; *Chan et al., 2013*). The AMOT BAR-like domain can remodel membranes in vitro (*Heller et al., 2010*), although it is not yet clear whether it functions as a classical or an inverse BAR domain. This distinction is mechanistically significant because classical BAR domains stabilize positive membrane curvature whereas inverse BAR domains stabilize negative curvature (*Mim and Unger, 2012*), both of which are generated during enveloped virus assembly.

In principle, the BAR domain could mediate AMOT p130 recruitment at the half-moon stage, once the nascent viral membrane has reached the proper degree of curvature. The BAR domain could also help drive the additional membrane curvature required to proceed to the lollipop stage. Similarly, F-actin recruitment and polymerization could help 'push' the virion toward the lollipop stage, although several recent studies argue strongly against an essential role for F-actin in HIV-1 assembly (*Bharat et al., 2014*; *Rahman et al., 2014*).

Our experiments indicate that another important AMOT p130 function is to recruit NEDD4L to sites of virus budding. In support of this idea, we have shown that: (1) the two proteins bind one another directly through interactions between the NEDD4L WW domains and PPXY elements present in the unique N-terminal region of the AMOT p130 isoform (*Figures 1, 2* and references *Wang et al., 2012*; *Moleirinho et al., 2014*), (2) AMOT p130 can recruit NEDD4L to cellular membranes, and this activity requires the NEDD4L WW-AMOT p130 PPXY interactions ([*Heller et al., 2010*] and our data not shown), (3) NEDD4L cannot stimulate release of 'crippled' HIV-1 constructs in the absence of AMOT (*Figure 4*), (4) NEDD4L mutants that cannot bind AMOT are also impaired in their ability to stimulate virus budding (*Figure 5*), (5) AMOT p80 and AMOT p130 PPXY mutants that cannot bind NEDD4L do not support virus budding (*Figures 3–6*) and (6) recombinant AMOT p130 and HIV-1 Gag proteins can bind one another directly, albeit weakly in vitro (*Figure 2B*), and could bind more robustly in cells owing to avidity and membrane topology effects.

Previous studies indicate that other retroviruses also employ distinct host factors to promote membrane envelopment and fission. The clearest example is the D-type Betaretrovirus Mason-Pfizer Monkey Virus (M-PMV), which forms spherical particles in the cytoplasm and then obtains its outer membrane and exits the cell by crossing the plasma membrane. M-PMV Gag contains two distinct late assembly domains that function non-redundantly: a PPPY motif that binds NEDD4 (and perhaps other NEDD4 family members), and a PSAP motif that binds TSG101/ESCRT-I (*Gottwein et al., 2003*). PPPY and PSAP mutations induce strikingly different phenotypes: mutation of the PPPY motif causes virions to stack up against the plasma membrane where they fail to initiate membrane envelopment whereas virions lacking the PSAP motif acquire an envelope, but accumulate at the lollipop stage or are released as chains of membrane-tethered particles. These observations imply that Gag PPPY-NEDD4 complexes are required for early stages of membrane bending and envelopment, whereas Gag PSAP-ESCRT-I complexes recruit the ESCRT machinery to mediate membrane fission. Like other family members, NEDD4 binds AMOT (*Moleirinho et al., 2014*), and we therefore speculate that M-PMV may also recruit AMOT to facilitate membrane bending and particle envelopment. This would imply that viruses can recruit NEDD4-motin complexes by binding either member of the complex. The deltaretrovirus HTLV-I is another case where a PPXY motif appears to be required for an early step in particle envelopment and a PTAP motif may function at a later membrane fission step ([*Le Blanc et al., 2002*; *Blot et al., 2004*; *Heidecker et al., 2004*], but also see reference [*Wang et al., 2004*]). Distinct roles for PPXY and PTAP motifs have not been delineated in other well-studied viruses that use both motifs, such as the gammaretrovirus Murine Leukemia Virus (*Yuan et al., 2000*) or the filovirus Ebola Virus (*Timmins et al., 2003*).

Although many details of HIV-1 assembly and budding remain to be elucidated, our data are consistent with a stepwise pathway in which: (1) AMOT p130 is recruited at the half-moon stage of HIV-1 assembly, perhaps by binding cooperatively to Gag and to appropriately curved membranes, (2) AMOT p130 promotes further membrane curvature and recruits NEDD4L via PPXY-WW domain interactions, (3) NEDD4L builds Lys-63-linked ubiquitin chains on the viral Gag protein (and/or other nearby substrates), (4) the juxtaposition of Gag late assembly domains and Lys-63-linked Ub chains creates high affinity binding sites for the early-acting ESCRT factors, TSG101 and ALIX, and (5) these factors then recruit the downstream ESCRT-III and VPS4 components required to effect membrane scission. This model is attractive because: (1) our work shows that AMOT functions upstream of the ESCRT pathway, (2) AMOT appears to function as a NEDD4L adaptor (see above), (3) ubiquitin transfer is required for efficient HIV-1 release (*Martin-Serrano, 2007*), (4) NEDD4L builds Lys-63-linked ubiquitin chains and can use HIV-1 Gag as a substrate (*Weiss et al., 2010*; *Sette et al., 2013*), and (5) both TSG101/ESCRT-I and ALIX have ubiquitin binding activities. The ALIX activity is specific for Lys-63-linked ubiquitin chains (*Dowlatshahi et al., 2012*; *Keren-Kaplan et al., 2013*; *Pashkova et al., 2013*), and ESCRT-I complexes also have multiple ubiquitin binding sites, although linkage specificity has not been observed (*Stefani et al., 2011*; *Agromayor et al., 2012*; *McCullough et al., 2013*). Cooperative binding to the late domains and ubiquitin could explain how ALIX and TSG101/

ESCRT-I can be efficiently recruited, despite making intrinsically weak interactions with the p6$^{Gag}$ late assembly domains. The proposed sequence of events could also provide a 'timing' mechanism for recruiting (or activating) these early-acting ESCRT proteins to function at a late stage of virion assembly, although the precise timing of early-acting ESCRT recruitment remains to be clarified because published reports differ on this issue (*Jouvenet et al., 2011*; *Ku et al., 2014*).

All enveloped viruses must bend and remodel membranes as they bud, and AMOT requirements may therefore be quite general. Indeed, AMOTL1 has been shown to bind the M protein of PIV5 and to be required for the efficient, ubiquitin-dependent release of this virus (*Pei et al., 2010*). Motin family members therefore contribute to the assembly of at least two highly diverged enveloped viruses; the paramyxovirus PIV5 and the lentivirus HIV-1. This suggests that AMOT proteins, like the ESCRT machinery, may provide general activities required for the assembly and release of a variety of different enveloped viruses.

## Materials and methods

### Plasmids, siRNA constructs and antibodies

Expression constructs, siRNA sequences, and antibodies are described in detail in *Supplementary file 1*. Constructs for expressing wild type and mutant NEDD4L and AMOT proteins in *E. coli*, insect SF-9 cells, and human 293T cells were created by standard cloning and mutagenesis methods, with detailed methods available upon request. All of the new expression constructs used in our studies have been deposited in the public DNASU plasmid repository (http://dnasu.org/DNASU/Home.jsp).

### Affinity purification and mass spectrometric identification of AMOT

For the experiment shown in *Figure 1B*, an empty vector control or a pCAG-OSF-NEDD4L expression vector (*Morita et al., 2007*) expressing NEDD4L with an N-terminal One-STrEP-FLAG (OSF) affinity tag was transfected into 293T cells (calcium phosphate method, 2 × 10 cm plates each, 25 µg DNA/plate). After 48 hr, cells from each sample were collected by scraping, pooled, washed three times with PBS (1.5 mM KH$_2$PO$_4$, 155 mM NaCl, 2.7 mM Na$_2$HPO$_4$, pH 7.4), and lysed with 500 µl Triton lysis buffer (1% Triton X-100, 50 mM Tris pH 8.0, 150 mM NaCl, 150 µg/ml PMSF). This, and all subsequent preparation steps, were performed at 4°C. Soluble extracts were pre-incubated for 30 min with 5 µl Protein-A resin slurry (Millipore, Temecula, CA) to remove non-specific binding proteins. Supernatants were collected and incubated for 2 hr with 10 µl STrEP-Tactin resin (IBA-Lifesciences, Göttingen, Germany). The resins were washed three times with 500 µl washing buffer (0.1% Triton-X100, 50 mM Tris, 150 mM NaCl, pH 8.0), resuspended in 35 µl SDS-PAGE loading buffer (2% SDS, 10% glycerol, 2% 2-mercaptoethanol (BME), 0.002% bromophenol blue and 62.5 mM Tris, pH 6.8), and boiled to release the bound proteins. 12 µl samples of each solution were electrophoresed (10% SDS-PAGE) and proteins were visualized by staining with Coomassie brilliant blue.

This procedure was repeated in three independent experiments, and in each case the protein band corresponding to AMOT p130 was visible in the Coomassie-stained gel (*Figure 1B*). This band was excised (and the equivalent region from the control lane was also excised in one of the repetitions), de-stained in 50% methanol with 50 mM ammonium bicarbonate (2 × 1 ml, 1 hr, 23°C, gentle vortexing), re-hydrated in 50 mM ammonium bicarbonate (1 ml, 30 min, 23°C), and cut into several pieces. Each piece was dehydrated in acetonitrile (1 ml, 30 min, 23°C, gentle shaking), and the excess acetonitrile was removed. Proteins were in-gel digested using 10–20 µl sequence-grade modified trypsin (Promega, Madison, WI, 20 ng/µl) in 50 mM ammonium bicarbonate (16 hr, 37°C), and the reaction was quenched by the addition of 20 µl 1% formic acid. The solution was removed and the gel washed twice with 50% acetonitrile in 1% formic acid (20 µl, 20 min, 37°C, with sonication) and once with 100% acetonitrile (20 min, 37°C, with sonication). The wash solutions were combined, dried, and reconstituted in 100 µl 5% acetonitrile with 0.1% formic acid for LC/MS/MS analysis.

Peptides were analyzed using a nano-LC/MS/MS system comprising a nano-LC pump (2D-ultra, Eksigent) and an LTQ-FT mass spectrometer (ThermoElectron Corporation, San Jose, CA), equipped with a nanospray ion source (ThermoElectron Corporation). 5–20 fmoles of each tryptic digest were injected onto a homemade C18 nanobore LC column (C18 [Atlantis, Waters Corporation, Milford, MA]); 3 µm particle; column (75 µm ID × 100 mm length), and eluted using a linear gradient of 5–70% solvent B in 78 min (solvent B: 80% acetonitrile with 0.1% formic acid; solvent A: 5% acetonitrile with 0.1% formic acid, 350 nl/min). The LTQ-FT mass spectrometer was operated in the data-dependent

acquisition mode (Xcalibur 1.4 software) with the 10 most intense peaks in each FT primary scan determined on-the-fly and trapped for MS/MS analysis and CID peptide fragmentation/sequencing in the LTQ linear ion trap. Spectra in the FT-ICR were acquired from $m/z$ 350 to 1400 at 50,000 resolution (∼2 ppm mass accuracy) with the following LTQ linear ion trap parameters: precursor activation time 30 ms and activation $Q$ at 0.25; collision energy at 35%; dynamic exclusion width at 0.1 Da–2.1 Da, with one repeat count and 10 s duration.

Raw LTQ FT MS data files were converted to peak lists with BioworksBrowser 3.2 software (ThermoElectron Corporation) using the following processing parameters: precursor mass 351–5500 Da; grouping enabled, 5 intermediate MS/MS scans; precursor mass tolerance 5 ppm, minimum ion count in MS/MS = 15, and minimum group count = 1. Resulting DTA files were merged, and searched to identify peptides/proteins against NCBInr using the MASCOT search engine (Matrix Science Ltd.; version 2.2.1, Boston, MA). Searches were done with tryptic specificity, allowing two missed cleavages, or 'non-specific cleavage' and a mass error tolerance of 5 ppm in MS spectra (i.e., FT-ICR data) and 0.5 Da for MS/MS ions (i.e., LTQ linear ion trap). Identified peptides were accepted when the MASCOT ion score value exceeded 20. In all three repetitions, at least two peptides corresponding to AMOT p130 were identified in the tryptic digest, and the maximal coverage was 33% (see *Figure 1—figure supplement 1*).

## Immunoprecipitations

To assay OSF-NEDD4L protein binding to endogenous AMOT p130 (*Figure 1C*), 293T cells ($2 \times 10^6$ cells per 10 cm plate) were transfected using Lipofectamine 2000 (Invitrogen/Life Technologies, Carlsbad, CA) with either 3.5 µg of an empty pCAG-OSF expression vector (control) or with pCAG-OSF vectors that expressed the designated OSF-NEDD4L proteins, using 3.5 µg of expression constructs for wild type NEDD4L or NEDD4L$_{\Delta WW}$, or 2 µg each of expression constructs for NEDD4L$_{C942A}$ or NEDD4L$_{\Delta WW/C942A}$. In this case, and in all other experiments, empty vector was added to keep total DNA levels constant (i.e., 1.5 µg empty pCAG-OSF vector in the case of NEDD4L$_{C942A}$ or NEDD4L$_{\Delta WW/C942A}$). 48 hr post transfection, cells were washed in PBS buffer and lysed (25 mM Tris, 150 mM NaCl, 1 mM EDTA, 0.5% CHAPS, pH 7.5, 10 min, 4°C). Soluble lysates were collected by centrifugation ($5000 \times g$, 5 min, 4°C) and incubated with 50 µl STrEP-Tactin resin (IBA-Lifesciences, Göttingen, Germany, 3 hr, 4°C) equilibrated with lysis buffer. The resin was washed (25 mM Tris-HCl, 150 mM NaCl, 1 mM EDTA, 0.1% CHAPS, pH 7.5, 3 × 1 ml), resuspended in 50 µl 2× SDS-PAGE loading buffer, boiled, and the released proteins were analyzed by SDS-PAGE and western blotting. Separated proteins were transferred to PVDF membranes, blocked with 5% non-fat dry milk for 45 min at RT, and incubated overnight at 4°C with primary antibodies (see *Supplementary file 1*). Secondary antibodies were anti-rabbit IgG or anti-mouse IgG conjugated to IRdye800 or IRdye700, respectively (1:10,000 and 1:20,000 respectively, Rockland Immunochemicals Inc., Gilbertsville, PA). All western blots were visualized using an Odyssey scanner (Li-Cor Biosciences, Lincoln, NB).

To assay OSF-NEDD4L protein binding to exogenous HA-AMOT p130 proteins (*Figure 1—figure supplement 2*), 293T cells ($2 \times 10^6$ cells per 10 cm plate) were co-transfected using Lipofectamine 2000 (Invitrogen/Life Technologies) with 3.5 µg of a pCAG-OSF vector expressing wild type OSF-NEDD4L and 3.5 µg of an empty pCAG vector (control), or pcDNA vectors expressing HA-AMOT p130 or HA-AMOT p130$_{\Delta PPXY}$. Thereafter, the co-immunoprecipitation protocol was identical to the one described above.

## Bacterial protein expression and purification

Proteins were expressed in the Rosetta-pLysS bacterial strain grown in auto-induction media ZYP-5052 (*Studier, 2005*).

### NEDD4L protein purification

*E. coli* expressing GST-NEDD4L proteins (2 l) were harvested, resuspended in 80 ml lysis buffer (50 mM Tris, 150 mM NaCl, 5% glycerol, 2 mM MgCl$_2$, 1 mM TCEP, 0.5% NP-40, 0.5 mM EDTA, pH 8.0) supplemented with the protease inhibitors PMSF (20 µg/ml), pepstatin (0.4 µg/ml), leupetin (0.8 µg/ml), and aprotinin (1.6 µg/ml). This and all subsequent steps were performed at 4°C. Lysate viscosity was reduced by addition of DNase (12.5 µg/ml, Roche, Basel, Switzerland). The lysate was clarified by centrifugation at $39,000 \times g$ for 45 min, and loaded onto 10 ml glutathione-superose resin (GE Healthcare Bio-Sciences, Pittsburgh, PA). The resin was washed

with high-salt wash buffer (20 mM Tris, 1 M LiCl, 0.01% NP-40, 1 mM TCEP, 0.5 mM EDTA, 5% glycerol, pH 8.0) followed by low-salt wash buffer (20 mM Tris, 150 mM NaCl, 1 mM TCEP, 0.5 mM EDTA, 5% glycerol, pH 8.0) prior to elution (50 mM Tris, 150 mM NaCl, 5% glycerol, 1 mM TCEP, 0.5 mM EDTA, 20 mM reduced glutathione pH 8.0). PreScission protease (4 nmol protease/120 nmol purified protein, GE Healthcare Bio-Sciences) was added to remove the GST affinity tag and the sample was dialyzed overnight against the same buffer without glutathione. The cleaved protein was further purified on a Q-column (10 HiTrap HP-Q, GE Healthcare Bio-Sciences) equilibrated in buffer A (20 mM Tris, 50 mM NaCl, 5% glycerol, 1 mM TCEP, 0.5 mM EDTA, pH 8.0), and eluted with a linear gradient to buffer B (Buffer A, but with 1 M NaCl). NEDD4L-containing fractions were pooled and dialyzed against storage buffer (20 mM Tris, 150 mM NaCl, 20% glycerol, 1 mM TCEP, pH 8.0) and stored at −80°C. Typical yields were 60 nmol/l culture, and the protein identities were confirmed by electrospray mass spectrometry (NEDD4L: $MW_{calc} = 98,472$ g/mol, $MW_{exp} = 98,474$ g/mol; NEDD4L$_{\Delta WW}$: $MW_{calc} = 98,012$ g/mol, $MW_{exp} = 98,014$ g/mol).

## Gag$_{\Delta p6}$ protein purification

*E. coli* expressing HIV-1$_{HXB2}$ Gag$_{\Delta p6}$-GST (4 l) were harvested and resuspended in 150 ml of lysis buffer (50 mM HEPES, 500 mM NaCl, 5% glycerol, 2 mM MgCl$_2$, 10 mM BME, 0.1% NP-40, 10 µM ZnCl$_2$, pH 7.5) supplemented with protease inhibitors (PMSF, pepstatin, leupetin, and aprotinin). This and all subsequent steps were performed at 4°C. Lysate viscosity was reduced by addition of DNase (Roche) and the lysate was clarified by centrifugation (39,000×*g*, 45 min). Nucleic acids were removed by adding polyethylenamine (PEI at pH 8.0) to a final concentration of 1% (wt/vol), followed by centrifugation (27,000×*g*, 20 min). The fusion protein was precipitated by addition of 0.35 equivalents of saturated ammonium sulfate (to 26% saturation). The resulting pellets were resuspended in 80 ml of GST Loading Buffer (50 mM HEPES, 500 mM NaCl, 5% glycerol, 10 mM BME, 0.01% NP-40 and 10 µM ZnCl$_2$, pH 7.5), and loaded onto 5 ml of glutathione-sepharose resin (GE Healthcare Bio-Sciences). The resin was washed with GST loading buffer, and the protein was cleaved overnight on-column by incubation in 80 ml cleavage buffer (50 mM Tris, 500 mM NaCl, 5% glycerol, 10 mM BME, and 10 µM ZnCl$_2$, pH 7.5) supplemented with 0.5 mg PreScission protease (GE Healthcare Bio-Sciences). The flowthrough containing the cleaved protein was diluted to lower the salt concentration to 200 mM, and loaded onto an SP-column equilibrated in buffer A (50 mM Tris, 50 mM NaCl, 5% glycerol, 10 mM BME, 10 µM ZnCl$_2$, pH 7.5). Gag$_{\Delta p6}$ was eluted with a linear gradient to buffer B (buffer A, but with 1 M NaCl). Protein-containing fractions were pooled and dialyzed against storage buffer (20 mM Tris, 500 mM NaCl, 20% glycerol, 10 mM BME, 10 µM ZnCl$_2$, pH 7.5) and stored at −80°C. Typical yields were 75 nmol/l, and the protein identity was confirmed by electrospray mass spectrometry ($MW_{calc, -Met1} = 50,762$ g/mol, $MW_{exp} = 50,758$ g/mol).

## Streptactin pull-downs of Gag$_{\Delta p6}$ and NEDD4L with OSF-AMOT

For the experiments described in *Figure 2* and *Figure 2—figure supplement 1*, SF9 cells (2 l) were infected at an MOI of 5 with Bac-to-Bac baculoviruses expressing OSF-tagged GFP (control), AMOT p130, AMOTL1 or AMOTL2. Pellets from 200 ml of culture were lysed in 10 ml of lysis buffer (1% NP-40, 150 mM NaCl, 20 mM Tris pH 8.0, 0.5 mM EDTA supplemented with protease inhibitors). All steps were performed at 4°C. Supernatants were collected by centrifugation for 30 min at 13,000 RPM in a microcentrifuge and 0.5 ml of each lysate was incubated with 30 µl of a slurry of STrEP-Tactin resin (1 hr). Resins were washed with high salt wash buffer (20 mM Tris, 1 M LiCl, 0.5 mM EDTA, 0.1% NP-40, pH 8.0), followed by low salt buffer (20 mM Tris, 150 mM NaCl, 0.5 mM EDTA, 0.1% NP-40, pH 8.0). For NEDD4L binding experiments, the OSF-GFP or OSF-AMOT-bound resins were washed with binding buffer (150 mM NaCl, 50 mM Tris, 1 mM TCEP, 10 µM ZnCl$_2$, 0.1% NP-40, pH 7.5). Pure recombinant NEDD4L or NEDD4L$_{\Delta WW}$ proteins (0.5 µM in 500 µl binding buffer) were pre-incubated with washed resins for 1 hr, and then incubated for 1 hr with GFP- or AMOT-bound resins. Resins were washed three times with 250 µl binding buffer, resuspended in 30 µl SDS-PAGE loading buffer, boiled, separated by SDS-PAGE, and visualized by Coomassie staining or by Western blotting (with 1 µl samples used for Coomassie staining, and 1 µl samples of a 50-fold dilution used for Western blotting).

For Gag$_{\Delta p6}$ binding experiments, the OSF-GFP (control)- or OSF-AMOT-bound resins were washed with binding buffer. Pure recombinant HIV-1 Gag$_{\Delta p6}$ (1 µM) in 500 µl binding buffer was

incubated with the washed resins for 1 hr. Resins were washed with binding buffer, and processed as described above.

## NEDD4L and AMOT co-overexpression experiments

NEDD4L and AMOT protein family co-overexpression experiments shown in *Figure 3* and *Figure 3—figure supplement 1*, *Figure 3—figure supplement 2*, and *Figure 7—figure supplement 1A* were performed following the protocol: 293T cells ($4 \times 10^5$ cells/well in a 6-well plate) were co-transfected (Lipofectamine, 2000; Invitrogen/Life Technologies) with 1 µg of an R9-based expression vector for the NL4-3 strain of HIV-1$_{\Delta PTAP, \Delta YP}$ (*Swingler et al., 1997*), 0.5 µg of pCI-FLAG NEDD4L or NEDD4L$_{\Delta WW}$ (*Chung et al., 2008*) or pCI-FLAG empty vector in combination with either 0.5 µg of pCMV AMOT p80 (*Figure 3—figure supplement 2*) or 1 µg pcDNA3 HA-AMOT p130 (*Figure 3* and *Figure 3—figure supplement 2*) or 0, 0.25, 0.5, 0.75, 1, 1.25 or 1.5 µg pcDNA3 HA-AMOT p130, (*Figure 3—figure supplement 1*) or 1 µg pcDNA3 HA AMOTL1 or 1 µg pcDNA3 HA AMOTL2 (*Figure 7—figure supplement 1A*). 48 hr post-transfection, media was harvested for titer measurements and virus purification, and cells were washed off the plate in PBS.

Viral titers were assayed in HeLa-TZM indicator cells using a β-galactosidase assay, and following the manufacturer's instructions (Promega, Madison, WI). Briefly, HeLa-TZM cells (5000 cells per well, 96-well plate) were infected with four different dilutions of virus-containing culture media, harvested after 48 hr, washed once (PBS, 4°C), and lysed in 25 mM Tris phosphate (pH 7.8), 2 mM DTT, 2 mM 1,2-diaminocyclohexane N,N,N′,N′-tetraacetic acid (DCTA), 10% glycerol and 1% Triton X-100 (10 min, 23°C). Cell lysates were incubated in 200 mM sodium phosphate buffer (pH 7.3), 2 mM MgCl$_2$, 100 mM BME and 1.33 mg/ml ortho-Nitrophenyl-β-galactoside for 60 min at 37°C. Reactions were terminated by addition of 200 µl 1 M Tris base and absorbance at 420 nm was read in a plate reader.

For western blot analyses, virions were pelleted by centrifugation through a 20% sucrose cushion (90 min, 15,000×g, 4°C) and resuspended in SDS-PAGE loading buffer. Cells were washed in PBS, lysed in RIPA buffer (10 mM Tris, 1 mM EDTA, 0.5 mM EGTA, 1% Triton X-100, 0.1% sodium deoxycholate, 0.1% SDS, 140 mM NaCl, pH 8.0) supplemented with protease inhibitors (cOmplete, Roche) (10 min, 4°C), and the insoluble material was removed by centrifugation (10 min, 15,000×g, 4°C). Primary antibodies and dilutions used for western blots are given in *Supplementary file 1*.

## AMOT siRNA depletion and re-expression experiments

Depletion/re-expression experiments (*Figure 4*, *Figure 4—figure supplement 1*, *Figure 4—figure supplement 2*, *Figure 5*, *Figure 7*, and *Figure 7—figure supplement 1*) were performed following the protocol: t = 0: 293T cells ($2.5 \times 10^5$ cells/well, 6-well plate) were transfected with 10 nM siRNA using 7.5 µl Lipofectamine RNAiMax (Invitrogen/Life Technologies); t = 24 hr: media changed and second siRNA co-transfection with 10 nM siRNA and 0.5 µg wild type HIV-1 R9 expression vector or 1 µg HIV-1$_{\Delta PTAP, \Delta YPXL}$ expression vector and 0.5 µg of pCI-FLAG NEDD4L (10 µl Lipofectamine, 2000; Invitrogen/Life Technologies); t = 72 hr: media and cells harvested for titer measurements and western blot analyses as described above. For the rescue experiments, siRNA-resistant expression constructs expressing AMOT p80, AMOT p130, AMOT p130$_{\Delta PPXY}$, AMOTL1 or AMOTL2 were co-transfected at t = 24 hr.

## AMOT expression and effects on HIV-1 release from HeLa and Jurkat T cells

AMOT mRNA is expressed in 293T cells and in white blood cells and lymphoid tissues, but cannot be detected in HeLa cells (*Moleirinho et al., 2014*). Western blots were performed to compare AMOT protein expression levels in 293T vs HeLa cells (*Figure 6—figure supplement 1*). 293T ($2 \times 10^6$ cells/10 cm plate) and HeLa M ($1.2 \times 10^6$ cells/10 cm plate) cells were seeded, harvested 72 hr later by trypsin treatment, washed twice in PBS, counted, lysed in RIPA buffer ($1.2 \times 10^5$ cells/µl) for 15 min at 4°C, clarified by centrifugation (15 min, 15,000×g, 4°C) and diluted 1:1 with 2× SDS loading buffer. 2.5, 5, 10 and 20 µl of each cell lysate mixture was loaded on a 4–15% gradient gel and AMOT proteins were detected by western blotting.

Experiments to test whether AMOT overexpression can stimulate HIV-1 release from HeLa cells (*Figure 6A*) were performed as follows: HeLa M cells ($2.5 \times 10^5$ cells/well in a 6-well plate) were co-transfected with 0.5 µg of wild-type R9 HIV-1 and 0.5, 1, 2 or 4 µg of pcDNA3-HA AMOT p130 expression vectors. Cells and viruses were harvested 48 hr post-transfection.

To test whether AMOT could stimulate virus release from physiologically relevant T cells, AMOT was overexpressed in Jurkat T cells as follows: t = 0, cells were harvested, washed in PBS and resuspended in Neon resuspension buffer R (Invitrogen/Life Technologies). $5 \times 10^6$ cells were combined with 3 μg of the wild type R9 HIV-1 expression vector and 0, 5 or 10 μg of pcDNA3-HA AMOT p130 or AMOT p130$_{\Delta PPXY}$ expression vectors. Cells were then electroporated in a 100 μl tip with the Neon Transfection System using three 10 ms pulses of 1325 V. Following electroporation, each sample was immediately transferred to a 10 cm plate containing 10 ml pre-warmed RPMI (supplemented with 10% FBS and 2 mM glutamine). t = 96 hr: media and cells were harvested.

Protocols for analyzing protein expression and viral titers were identical to those described for 293T cells experiments.

## Scanning electron microscopy (SEM)

293T cells ($2.5 \times 10^5$ cells/well, 6-well plate) were seeded onto glass cover slips (24 hr prior transfection) and co-transfected with the designated siRNAs and the R9 HIV-1 expression construct as described above. 48 hr post-transfection, the media was removed and cells were fixed in 1 ml fixation buffer (2.5% glutaraldehyde/1% paraformaldehyde in sodium cacodylate buffer; 50 mM sodium cacodylate, 18 mM sucrose, 2 mM CaCl$_2$, pH 7.4, 10 min). This and all subsequent steps were performed at 23°C. Fixed cells were washed three times in sodium cacodylate buffer (5 min), and stained with 2% OsO$_4$ for 1 hr. Each sample was then dehydrated in a graded ethanol series, dried in HDMS (hexamethyldisilazane), and sputter coated with platinum. SEM images were collected on a FEI Quanta 600 FEG scanning electron microscope at a beam energy of 10 kV, a spot size of three, and magnifications of 10,000, 30,000 or 65,000 X.

## Transmission electron microscopy (TEM)

293T cells ($2.5 \times 10^5$ cells/well, 6-well plate) were co-transfected with the designated siRNAs and the R9 HIV-1 expression construct as described above. 48 hr post-transfection, the media was removed and cells were fixed in 1 ml fixation buffer (2.5% glutaraldehyde/1% paraformaldehyde in sodium cacodylate buffer [50 mM sodium cacodylate, 18 mM sucrose, 2 mM CaCl$_2$, pH 7.4, 10 min]). This and all subsequent steps were performed at 23°C. Fixed cells were harvested, washed three times in sodium cacodylate buffer (5 min), and stained with 2% OsO$_4$ for 1 hr. The pellet was rinsed once in sodium cacodylate buffer and twice in water (5 min), followed by incubation in 100 μl of a 4% uranyl acetate solution (30 min). Stained cells were dehydrated in a graded ethanol series followed by acetone, and embedded in epoxy resin EMBed-812 (Electron Microscopy Sciences, Hatfield, PA). Thin sections (80–100 nm) were cut, post-stained with saturated uranyl acetate (20 min), rinsed with water, dried, stained with Reynolds' lead citrate (10 min) and dried again. TEM images were collected on a Hitachi H-7100 transmission electron microscope at an accelerating voltage of 75 kV, equipped with a Gatan Orius sc1000 camera.

Imaged virions were scored as 'mature' if they lacked an observable cellular tether and a condensed core was evident, as 'immature' if they were untethered but lacked a condensed core, and as 'budding' if a Gag layer was clearly evident and the nascent virion was assembling or was clearly tethered to the cell. The degree of assembly was assessed by rendering a (hypothetical) spherical particle and measuring the arc angle of the nascent Gag layer. To image 500 virions budding from wild type cells, it was necessary to screen about eight-times as many thin slices of wild type control cells (~2500 cells) vs TSG101- or AMOT-depleted cells (275 and 300, respesctively).

## Acknowledgements

We thank Chad Nelson and the University of Utah Mass Spectrometry Core facility for mass spectrometry analyses, Chris Rodesch and the University of Utah Cell Imaging Core facility for assistance with fluorescence microscopy, Nancy Chandler in the University of Utah EM Core facility for expert thin sectioning, S Joshua Romney (Invitrogen/Life Technologies) for the loan of a Neon transfection system, and members of our laboratory for critical reading and feedback. This work made use of University of Utah shared facilities of the Micron Technology Foundation Inc. Microscopy Suite sponsored by the College of Engineering, Health Sciences Center, Office of the Vice President for

Research, and the Utah Science Technology and Research (USTAR) initiative of the State of Utah. All new constructs reported in this manuscript have been submitted and are freely available from the DNASU repository (https://dnasu.org/DNASU/).

## Additional information

### Funding

| Funder | Grant reference number | Author |
|---|---|---|
| National Institute of Allergy and Infectious Diseases (NIAID) | AI051174 | Wesley I Sundquist |
| National Institute of General Medical Sciences (NIGMS) | GM082545 | Wesley I Sundquist |
| Japan Society for the Promotion of Science (JSPS) | | Jun Arii |

The funders had no role in study design, data collection and interpretation, or the decision to submit the work for publication.

### Author contributions

GM, SLA, Conception and design, Acquisition of data, Analysis and interpretation of data, Drafting or revising the article, Contributed unpublished essential data or reagents; JA, Conception and design, Acquisition of data, Drafting or revising the article; MSL, Conception and design, Drafting or revising the article, Contributed unpublished essential data or reagents; WIS, Conception and design, Analysis and interpretation of data, Drafting or revising the article

## Additional files

### Supplementary file

• Supplementary file 1. Expression plasmids, antibodies and siRNAs used in this study.

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
