## [Decision Letter]

Thank you for sending your work entitled “Angiomotin Functions in HIV-1 Assembly and Budding” for consideration at *eLife*. Your article has been favorably evaluated by Vivek Malhotra (Senior editor) and 3 reviewers, one of whom is a member of our Board of Reviewing Editors.

The Reviewing editor and the other reviewers discussed their comments before we reached this decision, and the Reviewing editor has assembled the following comments to help you prepare a revised submission.

It was shown in the manuscript that AMOT proteins can simultaneously interact with NEDD4l E3 ligase and HIV-Gag protein and thereby facilitate budding of newly formed HIV virions. The hypothesis is that Nedd4L could be indirectly recruited to Gag via these AMOT proteins. This is an attractive hypothesis because it would explain earlier findings that “crippled” HIV-1 Gag proteins lacking PTAP and YPXL late domains can be induced to function for budding by overexpression of Nedd4L, despite the lack of PPxY motifs in Gag, or any other obvious way to recruit Nedd4L to Gag protein. Functional budding assays demonstrate that AMOT and Nedd4L cooperate to stimulate budding of the crippled HIV-1 Gag, and that AMOT depletion reduces budding of wt HIV-1. Budding could be restored through re-expression of AMOT, AMOTL1, or AMOTL2 from plasmids. An extensive EM analysis of budding HIV-1 particles upon AMOT depletion was performed. AMOT depletion leads to a block in particle formation that is earlier (half-moon stage) than the one caused by Tsg101 depletion (lollipop stage).

The quality of the data presented here is high and the reviewers were enthusiastic about the overall importance of the presented data. However, the reviewers agreed that the following issues need to be addressed before the manuscript can be considered for publication in *eLife*:

More information on the physiological significance AMOT-Gag interactions should be provided. For example, AMOT does not exist in many cell types and most of the studies required a Gag missing the late domains. Maybe AMOT only plays a role in Gag budding missing the late domains – but never plays a role in normal physiological assembly. What is the importance of AMOT for virus budding in different cell types, and what is the status of HIV budding in HeLa cells that are naturally deficient for AMOT. Could AMOT be acting on something else which, in turn, facilitates assembly? Does AMOT play a role normally in assembly of HIV-1, or only when they have late domains missing? Is Amot functioning at the sites of assembly? What is the relative timing of AMOT and ESCRTs. What is the evidence that AMOT is a pre-requisite for recruitment of ESCRTs rather than being a parallel process? When AMOT is depleted in the HEK293 cells and the half-moon shapes are formed, is ESCRT recruited to these half-moon shapes?

The interaction of NEDD4 and AMOT has been already shown previously, where NEDD4 ubiquitination of AMOT leads to its degradation via UPS. Authors as well demonstrate in one figure that overexpression of NEDD4 leads to decrease in total level of AMOT. However, in contradiction to this, they used NEDD4 overexpression in all experiments. One would expect that presence of AMOT negatively affects HIV release, by blocking access of NEDD4 and that overexpression of NEDD4 hence neutralizes overexpression of AMOT, by targeting it for the UPS degradation. How is than degradation of AMOT regulation during the HIV budding? Does presence of Gag neutralizes NEDD4 mediated degradation of AMOT? Could authors test this in their assays, and correlate the levels of AMOT in setting where NEDD4 is not overexpressed, and HIV budding is followed? In such setting both wt and dPTAP Gag should be tested, and compared.

Does increased concentration of HIVGag in the binding assays can outcompete NEDD4 binding? Further, the domain/motif in AMOT that binds to HIVGag should be identified, and this mutant should be tested in all budding/infection assays.

Minor comments:

Reviewer #1:

1) Does HIVGag negatively influences NEDD4 mediated ubiqutination of AMOT? Could authors test this in cells or in vitro. In case is so, does that support the fact that AMOT is stabilized upon HIV infection and only than is not degraded but facilitates ubiquitination of Gag?

2) AMOT contains BAR domain, however authors do not have an attempt to test the requirement of BAR domain in proposed process, despite the fact that BAR domains are known to be able to remodel/recognize membrane curvature. Figure 3 shows clearly that overexpression of AMOT leads to formation of large aggregate-like structures that are rather artefacts of overexpression than functional co-localization of AMOT with NEDD4. Authors should look closely on localization of endogenous AMOT in cells, and demonstrate its co-localization with HIV-Gag. Picture of Hiv-Gag localization should be provided in both wt and AMOT deficient cells as well.

3) Figure 9 should contain pictures of wt and rescued cells as well, and not only TSG101 and AMOT depleted cells.

4) It is not clear how does presence of AMOT affect the assembly of ESCRT machinery via influencing NEDD4. Considering that AMOT deficient cells are not even able to form the buds one can speculate that AMOT might be important for the actin cytoskeleton regulation and membrane curvature around the bud. Could authors provide TIRF images, showing recruitment of AMOT to HIV buds, and subsequent recruitment of ESCRT components, as they propose in the Ddiscussion. Currently there is no evidence that AMOT acts upstream of ESCRT beside the fact that AMOT binds NEDD4. Does ESCRT get recruited at all in case of AMOT deficient cells? Does AMOT precede recruitment of ESCRT when budding events are followed by TIRF?

5) If AMOT is required for NEDD4 recruitment and presumably subsequent ubiquitination, could authors also provide western blots showing levels of HIVGag ubiquitination in case of AMOT depletion or overproduction, in setting where only endogenous NEDD4 is present?

6) Figure 8, right panel: more representative blot should be provided, since this one is not convincing at all. MA protein is not detected at all in viral particles, and the differences in CA level are almost not visible, going against the whole proposed mechanism that AMOT is able to surpass the lack of PTAP in crippled HIVGag.

7) Would be beneficial for the reader to have domain/motif comparison of all AMOTs (including AMOTL1 and AMOTL2) in Figure 1, or somewhere else in the figures. Based on which characteristic are those proteins redundant in terms of HIVGag budding.

Reviewer #2:

1) The binding experiments for the Nedd4L-AMOT interaction are carried out using a variety of constructs to localize binding determinants. Less so for the experiments characterizing the new interaction between Gag and AMOT. Have the authors tested whether AMOT p80 binds to Gag? If so, it would be useful to mention it, for example, on the third paragraph of the subsection “AMOT p130 is required for NEDD4L stimulation of HIV-1 budding”, in the Results section, referring to Figure 5–figure supplement 2: “Hence, Nedd4L stimulation… requires the presence of an AMOT p130 protein that can bind NEDD4L…”. The authors seem to attribute the lack of Nedd4L stimulation solely to lack of binding between p80 and NEDD4L. This is a case where it would be helpful to know if p80 binds to HIV-1 Gag. If not, it would mean that there are multiple defects in p80 that prevent it from functioning for the stimulation of Gag budding.

2) The sentence in the fourth paragraph of the subsection “AMOT p130 is required for NEDD4L stimulation of HIV-1 budding”, in the Results section, referring to Figure 5: “However, this stimulation was completely abrogated when endogenous AMOT proteins were depleted”. This is a key experiment in the paper. It is clear that full stimulation in these experiments requires both AMOT (endogenously expressed) and Nedd4L (from plasmid), and I believe that is the key point being made here. However, the statement that stimulation was completely abrogated when AMOTs were depleted may be a bit misleading. It implies that in the absence of AMOT, Nedd4L expression has no effect. But it does (compare lanes 2 and 4 of Figure 5). Please rephrase. This result is, perhaps, to be expected given that AMOTL1 and AMOTL2 are still present in these cells.

3) The sentence in the third paragraph of the subsection “AMOT p130 is required for NEDD4L stimulation 242 of HIV-1 budding”, in the Results section, referring to Figure 5–figure supplement 1: “the degree of rescue correlated positively with AMOT p130 re-expression levels…”. Certainly this appears to be true when looking at the infectivity assays in Figure 5–figure supplement 1, but such definitive language is not supported as well by the Gag budding experiment shown in the western blot of that figure.

4) The authors may want to point out that the partial rescue by AMOTL1 and AMOTL2 seen in Figure 8 could be related to differences in expression levels of the 3 AMOT proteins in the cells, which would be consistent with the dose dependence effect observed in Figure 7.

Reviewer #3:

It would help in the text to indicate which cells were used when (e.g., for the EM the cell type is not mentioned in the text nor the figure legend. It would help to have more information on the constructs used. The text says that HIV-1 was expressed using R9 wildtype. From looking back at a previous article that is referenced (Chung, 2008), it appears to be a pro-viral plasmid. However, that article does not have the R9 listed in its supplemental table of plasmids and references back to a paper from the year before (Fisher, 2007) that does not have the R9 listed. For Figures 4 and 5, there are panels called “virus”. Would it be better to call these “supernatant”? There are virions in both the cells and the supernatent. Why is there such a dramatic effect on infectivity when the effect on viral protein in the supernatant is not as pronounced?

It is claimed in the first paragraph of the subsection headed “AMOT binds directly to NEDD4L and to HIV-1 Gag2 in the Results section that NEDD4L binds AMOTp130 with at least 1:1 stoichiometry. How was this conclusion reached from the gel?

---

## [Author Response]

[Editors’ note: the authors asked for clarification about the revision requirements prior to resubmission.]

We have first listed your final set of requirements and our responses to them. In addition, we have also addressed many of the comments made in the original reviews, and we have also described those cases.

*Test the role of AMOT directly in physiologically relevant cells. If experiments with the T-cells are difficult, use macrophages instead*.

As the reviewers pointed out, our original manuscript did not test the role of AMOT in cells that are physiologically for HIV-1 infection. To address this shortcoming, we have now tested how altering AMOT p130 levels affects the release of infectious HIV-1 particles from Jurkat T cells. These experiments were performed by comparing the effects of overexpressing either wild type AMOT p130 or an inactive AMOT p130 mutant with point mutations in the three NEDD4-binding “PPXY” motifs (AMOT p130_ΔPPXY_). As shown in the new Figure 6, we find that overexpression of wild type AMOT p130 strongly stimulates HIV release and infectivity in a dose-dependent fashion, with maximal stimulations of 16-fold (virion release) and 7-fold (viral infectivity) vs. a transfected vector control. In contrast, overexpression of AMOT p130_ΔPPXY_ reproducibly reduced HIV-1 release and infectivity, presumably by dominantly inhibiting the functions of endogenous AMOT p130 proteins. We have confirmed that the levels of HIV-1 protein expression are comparable in all cases (not shown). AMOT expression levels in Jurkat T cells are beneath our detection limits, but we feel confident in concluding that AMOT stimulates HIV-1 release in this setting because the experiment is well controlled by the use of an inactive point mutant AMOT p130 construct, because the effects are dose dependent, and because we repeated the experiment three times with similar results.

In the process of investigating how best to perform this experiment, we also noticed that one of published the “global” siRNA screens for HIV cofactors was performed in Jurkat T cells (57). Gratifyingly, the data from that screen also indicate that AMOT is important for HIV-1 replication in Jurkat T cells because an shRNA that targeted AMOT protected Jurkat cells against the cytotoxicity that occurs upon HIV infection (although the functional role of AMOT was not pursued further). Thus, there are now two independent lines of evidence that AMOT p130 is important for HIV-1 replication in Jurkat T cells (and we have, of course, now referenced [57]).

*Explain why Tsg101 knockdown is so much more effective than AMOT in HIV infectivity*.

This is also a good point. We hypothesized that the strong, but incomplete effects that we were seeing might reflect incomplete silencing of endogenous AMOT p130. We therefore repeated the experiment with a new, freshly repurified batch of siRNA. Importantly, this new AMOT p130 siRNA depleted both AMOT p130 and p80 even more efficiently than our previous batch of siRNA (see new Figure 5, panel 3, lane 4). As shown in the new Figure 5, this siRNA treatment reduces viral titers by approximately 8-fold, and the inhibition of virion release is approaching the inhibition that we see for our best siRNA against TSG101 (right panels, compare lanes 3 and 4). We therefore conclude that most (or all) of the residual virus release and infectivity that we saw in previous experiments was due to incomplete silencing of AMOT p130 (and/or the residual presence of AMOT L1 and L2). These data indicate that AMOT p130 is (nearly) as important as TSG101 for HIV-1 release. Finally, we note that: 1) In addition to reducing viral budding, TSG101 depletion also reduces HIV-1 infectivity by inhibiting Gag processing and viral maturation, whereas AMOT depletion only appears to inhibit by blocking budding (see the data in Figure 9 and the discussion below), and 2) the level of infectivity seen upon depletion of either AMOT p130 or TSG101 is significantly greater than the infectivity reduction seen upon depletion of the third known HIV budding factor, ALIX (Figure 5).

Please note that we have not gone back and repeated all of our siRNA rescue experiments with the new batch of siRNA because this would have required repeating a very large number of experiments and because the new experiment did not change our original conclusions.

*Test the effect of AMOT expression with HIV-1 R-9 in Hela cells*.

This experiment, i.e., examining the release of wild type HIV-1_NL4-3_ from HeLa cells (expressed from the R9 proviral expression vector), was already shown in Figure 7 of the original paper (now Figure 6). As we have discussed, the misunderstanding arose from an error that we made in mislabeling the HIV-1 construct in the figure caption (for which we apologize) and also from an apparent misunderstanding of the nomenclature that we used within the figure (and elsewhere). Viruses and proteins labeled “HIV-1” and “Gag” without any subscripts refer to the wild type virus and Gag protein, respectively. Viruses and Gag proteins that lack late domains are labeled with the subscripts “ΔPTAP, ΔYP”. We apologize for the confusion, and note that AMOT expression strongly stimulates release of wild type HIV-1 in HeLa cells.

*More information on the physiological significance AMOT-Gag interactions should be provided*. *For example, AMOT does not exist in many cell types.*

Firstly, we note that AMOT is expressed in white blood cells and lymphoid tissues—the normal host cells for HIV-1 infection—and we have provided this information and relevant references in the manuscript. AMOT is not expressed in HeLa cells (but AMOTL2 is), and we show that exogenous expression of AMOT stimulates virus release from HeLa cells (Figure 6). AMOT is expressed in Jurkat T cells (and this point is now made explicitly and referenced in the text, but AMOT nevertheless appears to be limiting because exogenous AMOT expression also stimulates HIV release from Jurkat T cells (new Figure 6).

Reviewer #1:

Figure 3
*shows clearly that overexpression of AMOT leads to formation of large aggregate-like structures that are rather artefacts of overexpression than functional co-localization of AMOT with NEDD4*.

We agree (and had pointed that out ourselves), but the reviewer’s point is well taken and we have therefore removed this figure from the manuscript and simply mentioned the co-localization as “data not shown”. We think there is no doubt that AMOT p130 and NEDD4L bind one another in cells, and do so through interactions between the AMOT p130 PPXY motifs and the NEDD4L WW domains. We have shown this in several ways in our paper, including by co-IP (Figure 2). We have also referenced other relevant papers showing that motins bind NEDD4 family members.

Figure 9
*should contain pictures of wt and rescued cells as well, and not only TSG101 and AMOT depleted cells*.

We, and others, have previously published EM images of thin-sectioned cells that show budding of wild type HIV-1 particles. Wild type virions bud efficiently, however, and it is therefore difficult to obtain images from single thin sections that contain numerous examples of budding wild type virions. To help address this limitation, we have turned to scanning electron microscopy (SEM). The advantage of SEM is that we can look at the entire surface of a cell and therefore capture many more budding events in a single image. We have now included such images for virions budding from control cells and from cells depleted of TSG101 and AMOT (new Figure 8). These images make it clear that there is an enormous difference between the efficiency of virus budding from wild type cells vs. cells that lack either TSG101 or AMOT.

*Would be beneficial for the reader to have domain/motif comparison of all AMOTs (including AMOTL1 and AMOTL2) in*
Figure 1*, or somewhere else in the figures. Based on which characteristic are those proteins redundant in terms of HIVGag budding*.

We agree and have now included the requested domain comparison within the modified Figure 1.

Reviewer #2:

*The binding experiments for the Nedd4L-AMOT interaction are carried out using a variety of constructs to localize binding determinants. Less so for the experiments characterizing the new interaction between Gag and AMOT. Have the authors tested whether AMOT p80 binds to Gag? If so, it would be useful to mention it, for example, on the third paragraph of the subection “AMOT p130 is required for NEDD4L stimulation of HIV-1 budding”, in the Results section, referring to Figure 5–figure supplement 2: “Hence, Nedd4L stimulation… requires the presence of an AMOT p130 protein that can bind NEDD4L…”. The authors seem to attribute the lack of Nedd4L stimulation solely to lack of binding between p80 and NEDD4L. This is a case where it would be helpful to know if p80 binds to HIV-1 Gag. If not, it would mean that there are multiple defects in p80 that prevent it from functioning for the stimulation of Gag budding*.

We have not tested AMOT p80 binding to Gag because the p80 isoform does not express well and we have therefore been unable to make pure recombinant AMOT p80. The reviewer is therefore correct that there may be multiple reasons that the p80 isoform does not support virus release. We believe, however, that we have shown that the lack of NEDD4L binding is sufficient to explain the lack of function of AMOT p80. The paragraph that the reviewer refers to also describes an experiment in which we show that an AMOT p130 protein with substitution mutations in the three PPXY motifs also fails to support HIV budding (Figure 4, AMOT p130_ΔPPXY_, right panels, compare lanes 6 and 5) and we therefore feel justified in concluding that NEDD4L stimulation requires the presence of an AMOT p130 protein that can bind NEDD4L.

*The sentence in the fourth paragraph of the subection “AMOT p130 is required for NEDD4L stimulation of HIV-1 budding”, in the Results section, referring to*
Figure 5*: “However, this stimulation was completely abrogated when endogenous AMOT proteins were depleted”. This is a key experiment in the paper. It is clear that full stimulation in these experiments requires both AMOT (endogenously expressed) and Nedd4L (from plasmid), and I believe that is the key point being made here. However, the statement that stimulation was completely abrogated when AMOTs were depleted may be a bit misleading. It implies that in the absence of AMOT, Nedd4L expression has no effect. But it does (compare lanes 2 and 4 of*
Figure 5*). Please rephrase. This result is, perhaps, to be expected given that AMOTL1 and AMOTL2 are still present in these cells*.

The reviewer makes a good point. We have rephrased the sentence to read: “However, virus release was reduced to the levels seen in untreated control cells when endogenous AMOT proteins were simultaneously depleted (compare lanes 2-4).

*The sentence in the third paragraph of the subsection “AMOT p130 is required for NEDD4L stimulation 242 of HIV-1 budding”, in the Results section, referring to Figure 5–figure supplement 1: “the degree of rescue correlated positively with AMOT p130 re-expression levels…”. Certainly this appears to be true when looking at the infectivity assays in Figure 5–figure supplement 1, but such definitive language is not supported as well by the Gag budding experiment shown in the western blot of that figure*.

We take the reviewer’s point, and have modified the sentence as follows: “the degree of rescue generally correlated positively with AMOT p130 re-expression levels.”

*The authors may want to point out that the partial rescue by AMOTL1 and AMOTL2 seen in*
Figure 8
*could be related to differences in expression levels of the 3 AMOT proteins in the cells, which would be consistent with the dose dependence effect observed in*
Figure 7.

We agree, and have added the following sentence that addresses this point: “Neither AMOTL1 nor AMOTL2 was quite as effective as AMOT p130 in rescuing HIV-1 infectivity in this assay, possibly because these motins are intrinsically less active than AMOT p130 and/or because they were expressed at lower levels (left panel 2, compare lanes 3-5).”

Reviewer #3:

*It would help in the text to indicate which cells were used when (e.g., for the EM the cell type is not mentioned in the text nor the figure legend*.

We have now defined the cell type used in every experiment.

*It would help to have more information on the constructs used. The text says that HIV-1 was expressed using R9 wildtype. From looking back at a previous article that is referenced (Chung, 2008), it appears to be a pro-viral plasmid. However, that article does not have the R9 listed in its supplemental table of plasmids and references back to a paper from the year before (Fisher, 2007) that does not have the R9 listed*.

The reviewer is correct that R9 is a wild type proviral HIV-1 construct. In the previous version of the manuscript, there were two references for the relevant sentence of the experimental methods. One (48) referred to the first report of the R9 proviral plasmid, and the other (8) referred to our previous report of the NEDD4L plasmids used in this study. We have now separated the two references within the sentence to remove any confusion.

*For*
Figures 4 and 5*, there are panels called “virus”. Would it be better to call these “supernatant”? There are virions in both the cells and the supernatent*.

We understand the point, but prefer to use the term “virus” to remain consistent with other papers in the field.

Why is there such a dramatic effect on infectivity when the effect on viral protein in the supernatant is not as pronounced?

This is a good question. We (and others) generally observe a correlation between virion release and infectivity measurements, but the correlation is not perfect. This is why we believe it is important to measure both release and infectivity. One cause of these discrepancies is that budding defects can also induce defects in viral maturation and in virion contents. This is well characterized in the case of TSG101 depletion, where defective Gag processing and viral maturation contribute to loss of infectivity, even when virions are released. This effect is evident in the control experiment shown in Figure 9. Another effect that we have documented for TSG101 is that the released virions have reduced levels of reverse transcriptase. We believe that this is because even when virus budding does occur it is slow, and the processed RT proteins “float away” during the prolonged budding step. We envision that a similar effect probably occurs when AMOT depletion slows virus release, though we have not tested this directly. The important point is that aberrant kinetics of virion release appears to reduce infectivity, whereas any virions that eventually escape the cell will contribute to the signal in the release assay. Thus, treatments that disrupt (or restore) wild type virus budding kinetics can have differential effects on virion release and infectivity.

It is claimed in the first paragraph of the subsection headed “AMOT binds directly to NEDD4L and to HIV-1 Gag2, in the Results section, that NEDD4L binds AMOTp130 with at least 1:1 stoichiometry. How was this conclusion reached from the gel?

NEDD4L is a smaller protein than OSF-AMOT p130, yet the intensities of the Coomassie-stained NEDD4L band are significantly darker than the intensities of the OSF-AMOT p130 bands in the pulled down complexes shown in Figure 2, panel 1, lanes 9 and 10. Thus, there must be at least as many molecules of NEDD4L as AMOT p130 in the complexes.